# UrbanKGent: A Unified Large Language Model Agent Framework for Urban Knowledge Graph Construction

**Yansong Ning[1], Hao Liu[1,2]**[*]
[1] AI Thrust, The Hong Kong University of Science and Technology (Guangzhou)
[2] CSE, The Hong Kong University of Science and Technology
`yning092@connect.hkust-gz.edu.cn liuh@ust.hk`

## Abstract

Urban knowledge graph has recently worked as an emerging building block to distill critical knowledge from multi-sourced urban data for diverse urban application scenarios. Despite its promising benefits, urban knowledge graph construction (UrbanKGC) still heavily relies on manual effort, hindering its potential advancement. This paper presents **UrbanKGent**, a unified large language model agent framework, for urban knowledge graph construction. Specifically, we first construct the knowledgeable instruction set for UrbanKGC tasks (such as relational triplet extraction and knowledge graph completion) via heterogeneity-aware and geospatial-infused instruction generation. Moreover, we propose a tool-augmented iterative trajectory refinement module to enhance and refine the trajectories distilled from GPT-4. Through hybrid instruction fine-tuning with augmented trajectories on Llama 2 and Llama 3 family, we obtain UrbanKGC agent family[2], consisting of UrbanKGent-7/8/13B version. We perform a comprehensive evaluation on two real-world datasets using both human and GPT-4 self-evaluation. The experimental results demonstrate that UrbanKGent family can not only significantly outperform 31 baselines in UrbanKGC tasks, but also surpass the state-of-the-art LLM, GPT-4, by more than 10% with approximately 20 times lower cost. Compared with the existing benchmark, the UrbanKGent family could help construct an UrbanKG with hundreds of times richer relationships using only one-fifth of the data. Our data and code are available at https://github.com/usail-hkust/UrbanKGent.

## 1 Introduction

Urban Knowledge Graph (UrbanKG) aims to model intricate relationships and semantics within a city by extracting and organizing urban entities (e.g., POIs, road networks, etc.) into a multi-relational heterogeneous graph [1]. As an emerging building block, multi-sourced urban data are widely used to construct an UrbanKG to provide critical knowledge for various knowledge-enhanced urban downstream tasks, such as traffic management, pollution monitoring, and emergency response [2, 3, 4, 5]. UrbanKG has gradually become an essential tool of the modern smart city.

In prior literature, many efforts have been devoted to urban knowledge graph construction (UrbanKGC) using massive urban data sources. In particular, one commonly used approach [6, 7, 8] is to extract entities from structured urban data (e.g., geographic data, city sensor data, and traffic data) and define the relationships between obtained urban entities based on manually designed rules or patterns. However, these approaches suffer heavy reliance on a deep understanding of the application domain and are labor-intensive. Recently, inspired by the success of the Large Language Models (LLMs)

---

[*]Corresponding author.
[2]https://huggingface.co/usail-hkust/UrbanKGent-7B, https://huggingface.co/usail-hkust/UrbanKGent-8B and https://huggingface.co/usail-hkust/UrbanKGent-13B

38th Conference on Neural Information Processing Systems (NeurIPS 2024).

in knowledge graph construction [9, 10, 11], the LLMs have been adopted to improve UrbanKGC. For instance, GeoLM [12] is pretrained on crowdsourced geographical corpus for geospatial entity recognition and relation extraction. K2 [13] retrains Llama-2-7B model on manually processed and filtered geoscience text corpus for geospatial relation extraction. Nevertheless, these works rely on high quality **corpus annotation** and computational extensive **model retraining**, which may discourage researchers from adopting UrbanKG for their own work.

LLM agent [14, 15] has recently emerged and shown remarkable zero-shot capability for autonomous domain-specific task completion. For example, Voyager [16] is a LLM-powered agent for zero-shot game exploration without re-training, and LLMLight [17] is a traffic signal control agent with zero-shot LLM reasoning ability. These studies motivate us to construct tailored LLM agents to address the aforementioned limitations in UrbanKG construction.

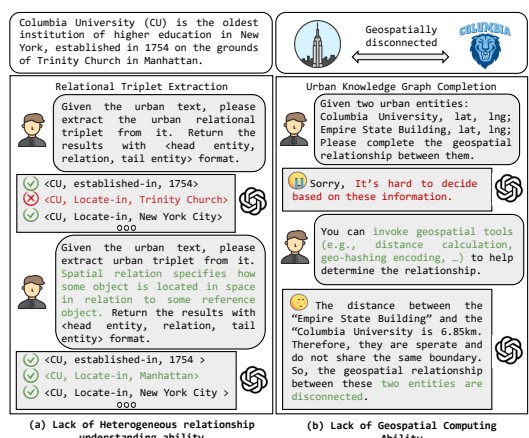

Figure 1: Illustrative example of urban relational triplet extraction and knowledge graph completion. (a) The heterogeneous relationship understanding limitation of LLMs can be addressed by injecting prior urban knowledge into instruction. (b) The geospatial computing limitation of LLMs can be alleviated by invoking external geospatial tools.

In fact, constructing an LLM agent compatible with various UrbanKGC tasks is a non-trivial problem due to the following two challenges: *(1) Challenges 1: How to adapt LLMs for UrbanKGC?* LLMs may not align well with the specific task due to the gap [18] between the natural language processing corpus for training LLMs and the domain-specific corpus in urban domain [19]. For example, the urban text data is usually heterogeneous and contains multifaceted urban knowledge (e.g., spatial, temporal, and functional aspects) [13]. As shown in Figure 1(a), the text description of *"Columbia University"* reflects its geographic spatial locations (i.e., spatial relationship), construction timelines (i.e., temporal relationship), and how it provides educational service for the city (i.e., functional relationship). LLMs may require a **tailored alignment to understand heterogeneous urban relationships** to extract these urban spatial, temporal, and functional relations accurately. *(2) Challenges 2: How to improve the capacity of LLMs for UrbanKGC?* The effectiveness of LLMs for urban knowledge graph construction is restricted by their feeble numerical computation capacity [20, 21], leading to their disability in complex geospatial relationship extraction [22, 23]. However, the urban geospatial relationship plays a vital role in urban semantic modeling [12] and has been widely incorporated in previous UrbanKGs [8, 24]. As can be seen in Figure 1, extracting *"disconnected"* relation between the geo-entity *"Columbia University"* and *"Empire State Building"* is useful for urban geo-semantic modeling. Accurately extracting such geospatial relationship demands necessary geospatial computing (e.g., utilizing latitude and longitude for distance calculation) and reasoning (i.e., deriving calculation results for geospatial relation reasoning) capabilities. It is appealing to improve the **geospatial computing and reasoning ability** of LLMs to satisfy the UrbanKGC task requirement.

To address the aforementioned challenges, in this study, we propose **UrbanKGent**, a unified LLM agent framework for automatic UrbanKG construction. Figure 2 illustrates the overview of UrbanKGent. For a given city, we first generate a knowledgeable instruction set for UrbanKGC tasks (relational triplet extraction and knowledge graph completion) from urban geographic and text data sources. By heterogeneity-aware and geospatial-infused instruction generation, as shown in Figure 1(a), various urban spatiotemporal relationship knowledge can be encoded into the instruction, which facilitates alignment between LLMs with the target UrbanKGC tasks. Moreover, we propose a tool-augmented iterative trajectory refinement module to enhance and refine the trajectory derived by distilling GPT-4 with the above constructed instructions. Based on geospatial tool augmentation and self-refinement, the deficiency of LLMs in geospatial computing and reasoning could be alleviated, and unfaithful trajectories could be filtered out. Finally, we perform hybrid instruction fine-tuning based on the enhanced and refined trajectories on Llama 2-7/13B and Llama 3-8B variants [25] by using LoRA [26]. The obtained UrbanKGent agent including 7/8/13B version, is feasible for completing multiple UrbanKGC tasks cost-effectively without extra GPT-API cost.

We conduct comprehensive experiments on two UrbanKGC tasks in two metropolises (New York City and Chicago) using both human evaluation and GPT-4-based self-evaluation. The empirical results validate the effectiveness of the proposed LLM agent for completing various UrbanKGC tasks. Moreover, the obtained UrbanKGent family (7/8/13B version) could help extract the same scale of triplets and entities of existing UrbanKG benchmark [8] using only one-fifth of data, and even expand the types of relations by hundreds of times.

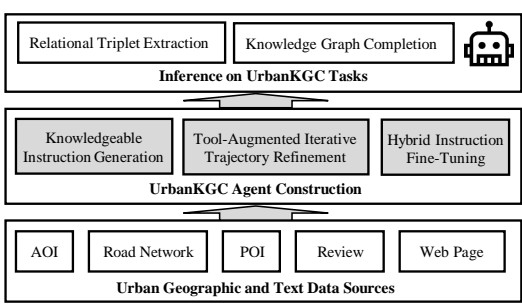

Figure 2: The framework of UrbanKGent.

Our contributions are summarized as follows: (1) We propose the first UrbanKGC agent framework **UrbanKGent** and UrbanKGent family to provide real-world UrbanKGC service, offering new opportunities to advance UrbanKG studies. (2) We propose a knowledgeable instruction generation module and a tool-augmented iterative trajectory refinement method, which align LLMs to UrbanKGC tasks and compensate for their geospatial computing and reasoning inability. (3) Extensive experiments on two real-world datasets validate the effectiveness of proposed framework and uncover its exceptional performance across UrbanKGC tasks.

## 2 UrbanKGC Data Description

### 2.1 Data Collection

We first acquire urban knowledge for two large cities New York City and Chicago from two data sources. Table 1 summarizes the statistics of the raw datasets.

#### 2.1.1 Geographic Data

The geographic data provides critical urban spatial structure information and functional semantics, which has been widely used in previous UrbanKG studies [8, 24, 27, 28].

**Area-Of-Interst (AOI) Data**. AOI data describes the urban spatial area structure, including urban commercial areas (e.g., shopping centers), residential areas (e.g., communities), and so on. In this work, we first follow UUKG [8] to acquire the AOI name and geometry value from NYC Gov [3] and CHI Gov [4]. Next, we use the AOI name to search their text description from Wikipedia and C4 [5] dataset. Each AOI record contains an AOI name, a polygon geometry value, and a text description. For example, *["Jamaica Bay", polygon (-73.86 40.58, ...), "Jamaica Bay is an estuary ..."]* is the record of the AOI *"Jamaica Bay"* with geometry value and text description.

**Road Network Data**. Road data describes the urban spatial network, including urban motorways, overpasses, and so on. We first follow [8] to obtain the road name and geometry value from Open Street Map (OSM [6]. Then, following the same text acquisition operation in AOI data, we crawl the textual description of each road record from Wikipedia. Each road record contains a road name, a linestring geometry value, a road type and a text description. For example, *["Central Park Avenue", linestring (-73.87 40.90, ...), primary, "Central Park Avenue is a boulevard in ..."]* describes the primary road named *"Central Park Avenue"* with a linestring geometry value and its textual description.

**Point-Of-Interest (POI) Data**. POI data represents different urban functions (e.g., residential and commercial), which have been widely adopted in many recent UrbanKG works [24, 6, 8]. We first follow [8] to obtain the POI name, and geometry value from OSM. Then the textual description of each POI record could be crawled from Wikipedia following the similar process. Each POI record contains a POI name, a coordinate geometry, a POI type, and a text description. For example,

---

[3]https://www.nyc.gov/

[4]https://www.chicago.gov/

[5]https://huggingface.co/datasets/allenai/c4

[6](https://www.openstreetmap.org/

*["Trump World Tower", coordinate (-73.96 40.75), residential, "Trump World Tower is a residential condominium ..."]* is the record of the POI *"Trump World Tower"*.

### 2.1.2 Text Data

The text data provides rich contextual knowledge of the city space from different perspectives (e.g., the spatial context) [13], and it plays an important role in geospatial understanding. In this work, we collect two types of text corpus.

Table 1: The statistics of raw datasets.

| Dataset | Description | New York City | Chicago |
|---|---|---|---|
| Geographic Data | # of AOI | 192 | 136 |
| | # of road | 6,765 | 2,241 |
| | # of POI | 5,872 | 5,877 |
| Text Data | # of review | 16,360 | 13,627 |
| | # of web page | 11,596 | 7,283 |

**Review Data**. The review of urban places provides commercial information that citizens use to make business decisions [29], playing a critical role in urban knowledge distillation. We collect review data from Google Map [7]. Specifically, we first manually split the city into multiple rectangular regions, then we utilize the Google Map API to query the places contained within each region and their reviews. Each review record contains a place name, a coordinate geometry value, a rating, and a text review. For example, *["Lifestyles Academy Inc", coordinate (-87.87 41.65), 4.9, "Very nice organization and ..."]* is the review record of place *"Lifestyles Academy Inc"*.

**Web Page Data**. The web page data works as the general text corpus for the city, and it contains rich geoscience knowledge that has been utilized in recent urban entity and relation extraction studies [13]. We collect web page data from the Google search engine. Specifically, we first input the name of the crawled AOI, Road, and POI record into Google. Then we concatenate the textual sentences of the top 10 retrieved web pages. Each web page record contains a long urban text description.

## 2.2 Data Preprocessing

Before constructing the UrbanKGC dataset, we first preprocess the raw datasets. We filter out AOIs, roads, POIs, reviews, and web pages whose crawled textual descriptions are null value, too short (e.g., less than ten word description) or meaningless (e.g., just repeating the POI name). In addition, we remove irrelevant information from the text description, such as non-English characters, non-ASCII gibberish, website addresses, and so on. More details can be found in Appendix A.

## 3 Preliminary

This section presents the UrbanKGC task definition and provides task analysis.

### 3.1 Task Definition and Problem Formulation

Before diving into the technical details, we first introduce the definition of UrbanKG:

**Definition 1** *UrbanKG. The UrbanKG is defined as a multi-relational graph $\mathcal{G} = (\mathcal{E}, \mathcal{R}, \mathcal{F})$, where $\mathcal{E}$, $\mathcal{R}$ and $\mathcal{F}$ is the set of urban entities, relations and facts, respectively. In particular, facts are defined as $\mathcal{F} = \{\langle h, r, t \rangle \mid h, t \in \mathcal{E}, r \in \mathcal{R}\}$, where each triplet $\langle h, r, t \rangle$ describes head entity $h$ is connected with tail entity $t$ via relation $r$.*

The UrbanKG encodes diverse urban semantic knowledge by connecting urban entities into a multi-relational graph. This work aims to construct an UrbanKG from collected unstructured text data. We decompose the UrbanKG construction (UrbanKGC) process into two sequential knowledge graph construction tasks, namely relational triple extraction [10] and knowledge graph completion [11]. We first provide the basic definition for these two subtasks, and then introduce the problem formulation of this work.

### 3.1.1 Task Definition

**Relational Triplet Extraction (RTE).** Given the unstructured texts, this task achieves joint extraction of entities and their relations [10] which are in the form of a triplet $\langle h, r, t \rangle$. For instance, given the

---

[7]https://www.google.com/maps

urban text sentence *"Columbia University is a private Ivy league research university in New York City."*, this task aims to identify two entities *"Columbia University"* and *"New York City"* and their relation *"locate-in"*, described as triplet *<Columbia University, locate-in, New York City>*.

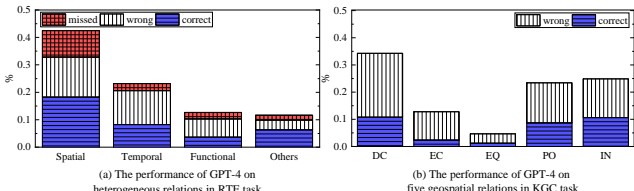

Figure 3: Quantitative performance analysis of prompting GPT-4 for UrbanKGC tasks. The result is obtained by comparing 50 GPT-4's outputs with the human's annotation.

**Knowledge Graph Completion (KGC).** Given a head entity $h$ and a tail entity $t$, this task is to predict the missing relation between them [11]. For instance, given the head entity *"Columbia University"* and the tail entity *"Empire State Building"*, this task is to predict that their missing relation, e.g., *"disconnected"*, described as triplet *<Columbia University, disconnected, Empire State Building>*.

### 3.1.2 Problem Formulation

Given the urban unstructured text data, the desired output is an UrbanKG $\mathcal{G}$. In this paper, this problem is decomposed into two sequential subtasks: (1) **Relational Triplet Extraction**: the first task extracts relational triplet $\langle h, r, t \rangle$ from the urban text data. The output of RTE task is $\mathcal{G}_1 = (\mathcal{E}, \mathcal{R}_1, \mathcal{F}_1)$, where $\mathcal{E}$ and $\mathcal{R}_1$ is the set of extracted entities and relations, while $\mathcal{F}_1$ is the set of extracted triplets. (2) **Knowledge Graph Completion**: for the given head entity $h$ and tail entity $t$ in $\mathcal{G}_1$, the second task is to predict the geospatial relationship[8] between them. The output of this task is $\mathcal{G}_2 = (\mathcal{E}, \mathcal{R}_2, \mathcal{F}_2)$, where $\mathcal{R}_2$ and $\mathcal{F}_2$ is the set of completed relations and triplets. By sequentially completing the above two tasks, we can obtain the constructed UrbanKG $\mathcal{G} = (\mathcal{E}, \mathcal{R}_1 \cup \mathcal{R}_2, \mathcal{F}_1 \cup \mathcal{F}_2)$.

## 3.2 Quantitative Task Analysis

As shown in Figure 1, we qualitatively find that LLMs lack urban heterogeneous relationship understanding ability and experience in geospatial computing and reasoning difficulty when adopting it for UrbanKGC tasks. This subsection presents a quantitative analysis of these two challenges.

**Heterogenous Relationship Understanding.** The ability to understand heterogeneous relationships is ubiquitous in distilling knowledge from the massive urban corpus. For example, the text description in Figure 1 illustrates a place from spatial location, temporal time, and functional aspects. Capturing these heterogeneous semantics is important for urban knowledge distillation. We perform quantitative analysis by randomly sampling 50 urban text data and then prompt GPT-4 to complete relational triplet extraction by providing only the basic task description. As shown in Figure 3(a), we find the LLMs experience serious misjudgment (i.e., extract wrong triplets or miss the triplet) on urban spatial, temporal, and functional triplet extraction. This indicates the limited capacity of LLMs to understand heterogeneous relationships.

**Geospatial Computing and Reasoning.** Geospatial computing and reasoning techniques are widely used in many previous UrbanKG studies [8, 24] for urban geospatial relation extraction. In recent works [23, 31], the geospatial skills of LLMs have also been demonstrated to lack geospatial awareness and reasoning ability [22]. To identify potential limitations, we quantitatively investigate how LLMs can perform on geospatial relation completion tasks. Specifically, we construct 100 head and tail entity pairs, covering five geospatial relations in the KGC task, and then prompt GPT-4 to predict with basic task description and geospatial relation candidates. As shown in Figure 3(b), we find that GPT-4 performs poorly on five geospatial relation completion. This further validates the disability of LLMs in geospatial computing and reasoning.

## 4 UrbanKGC Agent Construction

This section presents the proposed UrbanKGC agent construction framework.

---

[8]We follow GeoLM [12] to provide five RCC relationship [30] candidates: Disconnection (DC), external connection (EC), equality (EQ), partial overlap (PO), and tangential and non-tangential proper parts (IN). Details are in Appendix A.

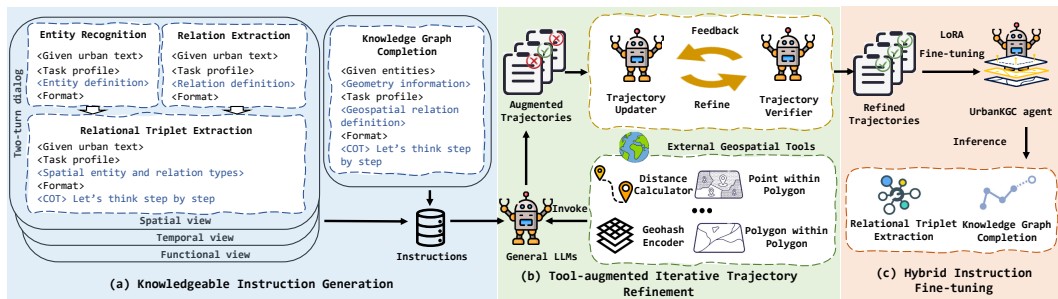

Figure 4: An overview of UrbanKGent Construction.

## 4.1 Overview

The overall pipeline of the UrbanKGent framework is illustrated in Figure 4. *(1) Knowledgeable Instruction Generation* consists of the heterogeneity-aware and geospatial-infused instruction generation modules for aligning LLMs to UrbanKGC tasks. *(2) Tool-augmented Iterative Trajectory Refinement* proposes geospatial tool interface invocation and iterative self-refinement mechanisms to enhance and refine generated trajectory. *(3) Hybrid Instruction Fine-tuning* fine-tune LLMs based on the refined trajectories for cost-effectively completing diverse UrbanKGC tasks.

## 4.2 Knowledgeable Instruction Generation

We first construct the knowledgeable instruction to adopt LLMs for two UrbanKGC tasks, including relational triplet extraction (RTE) and knowledge graph completion (KGC). Figure 4(a) illustrates the overview of the instruction construction process of these two tasks.

**Heterogeneity-aware Instruction Generation for Relational Triplet Extraction.** As discussed in Section 3, the urban text contains diverse heterogeneous relationships, thus we consider multiple views with both urban entity and relation definition for relational triplet extraction. In particular, we construct a multi-view instruction template for the urban relational triplet extraction, including spatial view, temporal view, and functional view. Each view is a multi-turn question-answer dialog [32] consisting of entity recognition, relation extraction, and triplet extraction module.

For the spatial view, we devise a two-turn dialog to align LLMs for spatial triplet extraction. In the first turn, we inject spatial entity and relation definition into the instruction template to guide LLMs to understand spatial characteristics and then extract potential spatial entities (e.g., *University*) and relations types (e.g., *locate-in*). In the second turn, the extracted types are explicitly fed into the instruction template for spatial triplet extraction. Intuitively, the spatial view allocates dedicated urban knowledge for LLMs to extract urban spatial relationships. Similarly, we construct the temporal view and functional view for corresponding temporal and functional triplet extraction, independently.

**Geospatial-infused Instruction Generation for Knowledge Graph Completion.** Despite heterogeneity-aware instruction enabling LLMs to extract urban triplets from various perspectives, the geospatial relationship between geospatial entities cannot be directly extracted. Therefore, we introduce a geospatial-infused instruction generation module to guide LLMs to complete missing geospatial relationships.

First, we incorporate the geometry information (i.e., the latitude and longitude) of geo-entities into instruction, so that the LLMs can utilize these geospatial values for relation inference. Second, we add the geospatial relationship definition to the instruction to guide LLMs in understanding the geospatial relationship definition. Intuitively, LLMs can refer to geospatial knowledge and make practical solutions for the KGC task. We provide the detailed instruction template in the Appendix B.

## 4.3 Tool-augmented Iterative Trajectory Refinement

### 4.3.1 Trajectory Generation

With the initial UrbanKGC instructions constructed, the following step is to generate reasoning trajectories [33], which will be used to fine-tune LLMs tailored to UrbanKGC task. Specifically, we follow FireAct [34] and use Chain-of-Thought (CoT) [35], a gradient-free technique, to prompt

GPT-4 (i.e., add prompt trigger *"Let's think step by step"* at the end of RTE and KGC instructions template) to generate the reasoning trajectories for UrbanKGC tasks.

The generated CoT trajectories could provide a step-by-step reasoning solution for UrbanKGC tasks. Nevertheless, the complex geospatial relationships cannot be easily extracted as discussed in Section 3 and recent geospatial reasoning works [31, 23, 22]. Therefore, we introduce a tool invocation module to guide LLMs to invoke tailored external geospatial tools [36] to enhance their geospatial computing and reasoning capacity for UrbanKGC tasks.

### 4.3.2 Tool Invocation for Trajectory Augmentation

We conduct two sequential procedures: tool invocation for geospatial computing support and trajectory deliberation for reasoning enhancement.

**Tool Invocation.** First, we construct a geospatial reasoning toolkit (e.g., distance calculation, eight interfaces in total shown in Table 6) by prompting GPT-4 for self-programming. Then, we construct tailored prompts to guide LLMs to invoke these interfaces. Specifically, the prompt is concatenated with an illustrative description of the function of each geospatial tool and a task instruction (i.e., *"Which types of tool interface you need"*). Intuitively, the external tool allocates calculation results for LLMs to infer missing geospatial relation. The toolkit description can be found in Appendix C.

**Trajectory Deliberation.** After manipulation with external tools, we prompt LLMs to refine uncertain reasoning steps based on these obtained manipulation results. Specifically, we construct the prompt by concatenating with manipulation results (e.g., the distance and geohash value of geo-entity) and a task instruction (i.e., *"Please refine your reasoning process"*). After feeding the prompt into GPT-4, the enhanced trajectory is obtained. Detailed prompt information can be found in Appendix C.

### 4.3.3 Iterative Trajectory Self-refinement

Despite tool-augmented deliberation improving the geospatial computing and reasoning ability of LLMs, enhanced trajectories may not all be faithful [37]. To alleviate potential error and ensure the trajectory quality [38], we refine these trajectories via an iterative self-refinement mechanism [39]. Specifically, we iterate two sequential blocks: (i) Trajectory verifier: given the trajectory, the verifier aims to provide feedback for refining the reasoning process; (ii) Trajectory updater: given the trajectory and feedback, the updater will further refine the current trajectory based on the feedback.

**Trajectory Verifier.** We construct a tailored prompt to ask LLMs to generate feedback. Specifically, we use a simple but effective trigger (*"Judge whether all extracted triplets are correct and provide improvement suggestion"*) to prompt LLMs to provide feedback. If the trajectory no longer requires modification, we let LLMs respond with *"This is a faithful trajectory"*. Such a verification step lets LLMs make reflections and improve the correctness of the trajectory.

**Trajectory Updater.** The updater utilizes provided feedback to refine the current trajectory via prompt trigger *"Follow suggestion to refine the reasoning process"*. Intuitively, the feedback may address multiple aspects (e.g., missed triplet in the RTE task or unfaithful reasoning process in the KGC task) of the unfaithful trajectories.

We iterate the trajectory verifier and updater until the predefined stopping condition is satisfied. The stopping condition is determined by either meeting the maximum number of iterations (we set it at three to avoid excessive cost) or when the verifier confirms all trajectories are faithful. Upon meeting the stopping condition, we use the last refined trajectory for further fine-tuning. Detailed prompt information can be found in Appendix C.

### 4.4 Hybrid Instruction Fine-Tuning

To construct a cost-effective UrbanKGC agent, we further utilize trajectories (generated by GPT-4) to fine-tune a smaller open-source LLM for faster inference speed and lower cost (i.e., prompting GPT-4 for UrbanKGC is expensive). Specifically, we finetune the LLM via the mixed-task instruction-tuning strategy [33]. The goal is to enhance the LLMs' capabilities in diverse UrbanKGC tasks.

**Mixture Training.** Set the base language model as $\mathcal{M}$, and $\mathcal{P}_{\mathcal{M}}(y \mid x)$ represents the probability distribution of response $y$ given instruction $x$. We consider the trajectory set on two UrbanKGC tasks, i.e., $\mathcal{D}_{RTE}$ and $\mathcal{D}_{KGC}$. Since both the instruction and the target output are formatted in natural

Table 2: The statistics of constructed UrbanKGC dataset.

| Dataset | | NYC-Instruct | NYC | NYC-Large | CHI-Instruct | CHI | CHI-Large |
|---|---|---|---|---|---|---|---|
| Records | RTE | 232 | 2,089 | 40,480 | 122 | 1,102 | 28,868 |
| | KGC | 232 | 2,080 | 33,534 | 122 | 1,101 | 28,607 |

language, we can unify the training into an end-to-end sequence-to-sequence way. Formally, the optimization process aims to minimize the loss of language model $\mathcal{M}$ as follows:

$$\mathcal{L} = \mathbb{E}_{(x,y)\sim\mathcal{D}_{\text{RTE}}} \left[\log \mathcal{P}_{\mathcal{M}}(y \mid x)\right] + \mathbb{E}_{(x,y)\sim\mathcal{D}_{\text{KGC}}} \left[\log \mathcal{P}_{\mathcal{M}}(y \mid x)\right], \qquad (1)$$

where $x$ and $y$ represent the instruction input and instruction output in the trajectory, respectively.

**Training Setup.** We choose the chat version of open-sourced Llama 2-7/13B and Llama-3-8B as our backbone models, and fine-tune Llama using LoRA strategy [26].

### 4.5 Inference on UrbanKGC Task

Via hybrid instruction fine-tuning, the obtained LLM UrbanKGent can be trained to follow the instructions to finish the UrbanKGC task. We prompt UrbanKGent to complete UrbanKGC tasks by following the pipeline shown in Figure 4. For the RTE task, we sequentially execute entity recognition, relation extraction, and relational triplet instruction generation, iterative self-refinement and output the extracted triplets. For the KGC task, we sequentially execute KGC instruction generation, external tool augmentation, iterative self-refinement block, and finally output the completed triplets.

## 5 Experiments

### 5.1 Experimental Settings

**Dataset.** In this work, two sequential tasks (i.e., RTE and KGC) of UrbanKGC are within an open-world setting (i.e., no predefined ontology) [40, 41]. We construct the RTE and KGC datasets of NYC and CHI by sampling uniformly from five raw data in Table 1, respectively. As shown in Table 2, we first construct two small datasets (i.e., NYC-Instruct and CHI-Instruct) for instruction fine-tuning and two middle datasets (i.e., NYC and CHI) to validate the performance of the constructed UrbanKGC agent. The remaining data works as the large-scale UrbanKGC dataset (i.e., NYC-Large and CHI-Large) in real-world scenarios shown in Table 5. The three types of datasets are non-overlapping to prevent data leakage. More dataset construction details are in Appendix A.

**Baseline Methods.** We provide a comprehensive comparison of our method with existing paradigms: **(1) End-to-end Models:** For the zero-shot RTE task, we utilize the end-to-end generation model RelationPrompt [42] and PRGC [10]. For the KGC task, we fine-tune KG-BERT [43] and KG-T5 [44] with the QA pairs constructed from the self-instruct dataset. **(2) LLMs-based Zero-shot Reasoning [45]:** We directly prompt the LLMs with basic task definitions to get the answer without training. **(3) LLMs-based In-context Learning [35]:** We sample 3-shot QA pairs as demonstrations from the self-instruct dataset as examples and get the answers from the LLMs without training. **(4) Vanilla Fine-tuning [11]:** We directly fine-tune the LLMs using the QA pairs constructed from the self-instruct dataset, and then prompt the LLMs with basic task definition without demonstrations. **(5) UrbanKGent Inference:** We directly prompt the LLMs using the UrbanKGgent inference pipeline in Section 4.5. The prompt templates of the above baseline methods are in Appendix B.

**Implementation and Detail Settings.** In our experiment, we select Vicuna [46], Alpaca [47], Mistrial [48], Llama-2 [49], Llama-3 [50], GPT-3.5 [51] and GPT-4 [51] as the backbone LLM $\mathcal{M}$. All experiments are conducted on eight NVIDIA A800 GPUs. For the GPT-3.5 and GPT-4, we adopt the gpt-3.5-turbo-16k-0613 API and gpt-4-0613 API.

**Evaluation Protocol.** Since UrbanKGC tasks in this work follow an open-world setting where labels are not visible, the classical metric (e.g., F1 and Hits@10) is not applicable. In this work, we regard evaluation as the binary classification, i.e., if the extracted triplet in RTE task is correct and if the completed relation in KGC task is correct. We follow recent LLMs-based KGC works [11] to employ accuracy as an evaluation metric. To make a comprehensive evaluation of the experimental results, we employ both of the human evaluation and GPT evaluation, which has been widely used in many LLM studies [52, 45]. For **Human Evaluation**, we employ human annotators to evaluate the results on

Table 3: The main results of relational triplet extraction (RTE) and knowledge graph completion (KGC). We report the accuracy (acc) and confidence for GPT evaluation on two datasets, and report accuracy (acc) for the Human evaluation approach. The best baseline performance is underlined.

| Type | Models | NYC GPT (acc/confidence) RTE | KGC | Human (acc) RTE | KGC | CHI GPT (acc/confidence) RTE | KGC | Human (acc) RTE | KGC |
|---|---|---|---|---|---|---|---|---|---|
| End-to-end Models | KG-BERT | - | 0.24/3.15 | - | 0.23 | - | 0.19/4.12 | - | 0.24 |
| | KG-T5 | - | 0.21/4.02 | - | 0.21 | - | 0.15/3.98 | - | 0.24 |
| | RelationPrompt | 0.12/3.38 | - | 0.12 | - | 0.21/3.53 | - | 0.18 | - |
| | PRGC | 0.08/4.01 | - | 0.06 | - | 0.13/4.15 | - | 0.15 | - |
| Zero-shot Reasoning | Vicuna-7B | 0.12/2.84 | 0.19/4.06 | 0.14 | 0.16 | 0.22/4.12 | 0.14/4.03 | 0.21 | 0.18 |
| | Alpaca-7B | 0.11/3.75 | 0.17/3.87 | 0.15 | 0.17 | 0.23/3.96 | 0.16/4.15 | 0.20 | 0.16 |
| | Mistral-7B | 0.14/4.12 | 0.21/4.11 | 0.17 | 0.18 | 0.21/3.75 | 0.15/3.76 | 0.19 | 0.19 |
| | Llama-2-7B | 0.14/1.98 | 0.18/3.75 | 0.16 | 0.18 | 0.26/1.96 | 0.15/2.83 | 0.21 | 0.22 |
| | Llama-3-8B | 0.15/4.02 | 0.15/4.02 | 0.20 | 0.21 | 0.24/3.75 | 0.15/4.08 | 0.22 | 0.22 |
| | Llama-2-13B | 0.21/2.07 | 0.28/3.91 | 0.19 | 0.22 | 0.22/2.19 | 0.16/2.47 | 0.22 | 0.24 |
| | Llama-2-70B | 0.25/3.07 | 0.28/3.75 | 0.22 | 0.24 | 0.27/3.55 | 0.16/2.47 | 0.24 | 0.23 |
| | Llama-3-70B | 0.24/4.18 | 0.29/4.31 | 0.23 | 0.24 | 0.26/3.98 | 0.17/4.26 | 0.25 | 0.23 |
| | GPT-3.5 | 0.29/4.11 | 0.36/3.47 | 0.31 | 0.23 | 0.31/3.79 | 0.31/3.16 | 0.31 | 0.29 |
| | GPT-4 | 0.38/4.03 | 0.39/3.82 | 0.41 | 0.29 | 0.39/4.08 | 0.32/4.03 | 0.43 | 0.35 |
| In-context Learning | Llama-2-7B | 0.18/2.15 | 0.21/3.96 | 0.19 | 0.18 | 0.25/2.44 | 0.18/3.27 | 0.23 | 0.20 |
| | Llama-3-8B | 0.17/4.06 | 0.18/3.53 | 0.21 | 0.22 | 0.28/4.31 | 0.17/4.14 | 0.24 | 0.21 |
| | Llama-2-13B | 0.26/3.52 | 0.31/3.28 | 0.23 | 0.24 | 0.28/2.65 | 0.21/2.53 | 0.25 | 0.26 |
| | GPT-3.5 | 0.41/4.65 | 0.42/4.08 | 0.42 | 0.31 | 0.36/4.24 | 0.36/4.23 | 0.39 | 0.36 |
| Vanilla Fine-tuning | Llama-2-7B | 0.32/4.37 | 0.38/3.65 | 0.32 | 0.27 | 0.29/3.80 | 0.30/3.65 | 0.33 | 0.31 |
| | Llama-3-8B | 0.31/4.18 | 0.35/4.18 | 0.35 | 0.26 | 0.31/4.18 | 0.29/4.15 | 0.32 | 0.34 |
| | Llama-2-13B | 0.35/4.26 | 0.41/3.92 | 0.39 | 0.29 | 0.31/4.14 | 0.29/3.87 | 0.37 | 0.35 |
| UrbanKGent Inference | Vicuna-7B | 0.24/3.07 | 0.24/3.95 | 0.29 | 0.23 | 0.27/4.12 | 0.22/3.95 | 0.23 | 0.25 |
| | Alpaca-7B | 0.26/3.85 | 0.27/3.83 | 0.26 | 0.22 | 0.27/3.83 | 0.21/4.12 | 0.27 | 0.29 |
| | Mistral-7B | 0.26/4.15 | 0.25/4.08 | 0.28 | 0.23 | 0.25/3.61 | 0.21/4.08 | 0.25 | 0.26 |
| | Llama-2-7B | 0.27/3.05 | 0.26/4.12 | 0.28 | 0.24 | 0.27/2.87 | 0.24/3.54 | 0.26 | 0.29 |
| | Llama-3-8B | 0.29/4.15 | 0.31/4.08 | 0.33 | 0.26 | 0.26/3.28 | 0.24/3.97 | 0.30 | 0.31 |
| | Llama-2-13B | 0.31/3.87 | 0.32/3.56 | 0.35 | 0.27 | 0.28/3.24 | 0.26/3.28 | 0.31 | 0.32 |
| | Llama-2-70B | 0.33/4.28 | 0.35/4.27 | 0.33 | 0.29 | 0.29/3.80 | 0.28/4.01 | 0.32 | 0.34 |
| | Llama-3-70B | 0.35/4.26 | 0.36/4.81 | 0.34 | 0.28 | 0.29/4.12 | 0.29/4.81 | 0.31 | 0.35 |
| | GPT-3.5 | 0.43/4.12 | 0.46/3.88 | 0.43 | 0.34 | 0.40/4.21 | 0.39/3.87 | 0.46 | 0.41 |
| | GPT-4 | 0.45/4.08 | 0.48/4.02 | 0.47 | 0.42 | 0.46/4.17 | 0.41/4.35 | 0.52 | 0.43 |
| UrbanKGent-7B | | 0.46/4.12 ↑2.22% | 0.49/3.97 ↑2.08% | 0.48 ↑2.08% | 0.44 ↑4.76% | 0.49/4.28 ↑6.52% | 0.43/4.58 ↑4.88% | 0.54 ↑3.84% | 0.45 ↑4.66% |
| UrbanKGent-8B | | 0.47/3.97 ↑4.44% | 0.51/4.15 ↑6.25% | 0.49 ↑4.26% | 0.45 ↑7.14% | 0.49/3.97 ↑6.52% | 0.44/4.05 ↑7.32% | 0.55 ↑5.77% | 0.46 ↑6.98% |
| UrbanKGent-13B | | 0.52/4.38 ↑15.56% | 0.56/4.13 ↑14.29% | 0.54 ↑14.89% | 0.47 ↑11.90% | 0.53/4.15 ↑15.22% | 0.48/4.42 ↑17.07% | 0.59 ↑13.46% | 0.49 ↑13.95% |

200 random samples. As for the **GPT Evaluation**, we use GPT-4 to evaluate the model performance on the full data to escape intensive labor. In this work, the GPT-4's evaluation has been demonstrated to be consistent with the human evaluation. Detail is in Appendix D.

## 5.2 Main Result

Table 4: Statistics comparison of constructed UrbanKGs in New York and Chicago between UrbanKGent and existing benchmark.

| Dataset | # Entity | # Relation | # Triplet | Data Volume |
|---|---|---|---|---|
| NYC-Large | 228,928 | 2,138 | 905,442 | 40,480 |
| CHI-Large | 95,813 | 1,336 | 563,290 | 28,607 |
| NYC-UUKG | 236,287 | 13 | 930,240 | 236,277 |
| CHI-UUKG | 140,602 | 13 | 564,400 | 140,577 |

The performance results are reported in Table 3. As can be seen, the constructed agent outperforms all thirty-one baseline models on two UrbanKGC datasets. Specifically, the UrbanKGent-13B achieves (15.56%, 14.29%, 14.89%, and 11.90%) improvements compared with the state-of-the-art GPT-4 with the same inference pipeline on NYC. The improvements on CHI are (15.22%, 17.07%, 13.46%, and 13.95%), respectively. Moreover, the UrbanKGent-7/8B also achieve comparable performance compared with the GPT-4.

Meanwhile, we observe that the zero-shot LLMs perform poorly in the UrbanKGC tasks, even using GPT-4. In addition, although the demonstrations provided by In-context-learning can incorporate the UrbanKGC task information, the performance gain is limited. Besides, we find that fine-tuning LLMs can make obvious improvements in the overall performance. Through vanilla fine-tuning, the Llama-2-7/13B and Llama-3-8B could achieve comparable performance with GPT-3.5 under the ZSL settings.

Moreover, although the various LLM backbones using the UrbanKGent inference pipeline perform slightly worse than the vanilla fine-tuning method, they could obtain better performance compared

with zero-shot reasoning and In-context learning paradigms. Such results demonstrate the benefit of knowledgeable instruction design and external tool innovation, but also indicate its performance bottleneck. As a deeper exploration, our work fills this gap through hybrid instruction fine-tuning, and the fine-tuned UrbanKGC agents, whether 7B, 8B or 13B, can achieve state-of-the-art performance in UrbanKGC tasks. We provide an in-depth analysis of the proposed UrbanKGent framework in Appdendix E.2.

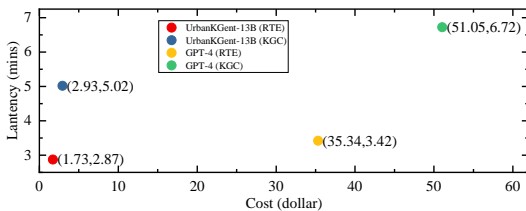

Figure 5: The model latency and cost of constructed UrbanKGent-13B and GPT-4 in UrbanKGC. We report the total inference time and cost of 1,000 RTE and KGC tasks.

### 5.3 Agent Application

We first derive UrbanKGent-13B for initial UrbanKGs acquisition in New York City and Chicago. After proper filtering and merging of the triplets, we obtain two large-scale UrbanKGs shown in Table 4. Compared with existing UrbanKG benchmark [8], we only use roughly one-fifth of the data for constructing the UrbanKGs with the same scale of triplets and entities, and even expanding the variety of relationships to a hundred times the original types. Moreover, we also provide efficiency analysis in Figure 5. As can be seen, UrbanKGent-13B achieves lower inference speed in latency and reduce the cost by roughly 20 times in both of RTE and KGC tasks. More details is in Appendix E.3.

## 6 Related work

**Domain-Oriented Agent Construction.** The concept of language agent [34] has become very popular recently, and a variety of LLM agents targeting different domains have been proposed. For example, Voyager [16] is constructed for automated game exploration, WebGPT [17] is an HTML agent for diverse document understanding tasks, LLMLight [53] constructs a language agent for transportation domain, K2 [13], GeoGalactica [19] and GeoLLM [12] propose to re-train language agent for geospatial semantic understanding. In addition, many recent works like Auto-GPT [54] and CAMEL [55] aim at proposing an autonomous agent framework for agent construction. Nevertheless, there is still no UrbanKGC agent construction framework for the urban computing domain.

**LLMs for Knowledge Graph Construction.** Recently, the advent of LLMs [56] invigorated the field of NLP. Many studies have begun to explore the potential of LLMs in the domain of KG construction. For example, [32, 57] finds that transforming the NER and RE task into a multi-turn question-answering dialog could improve the model performance. [9] explicitly derive syntactic knowledge to guide LLMs to think, which could develop the performance of NER. Despite these LLM-driven KG construction methods [58, 40] in general domains being widely investigated, KG construction in urban domain still remains an open challenge [59].

**Urban Knowledge Graph.** Urban knowledge graph has been proven useful in various urban tasks, such as traffic flow prediction [60, 61, 27, 62], mobility prediction [6], site selection [7], city profiling [63], crime prediction and so on [8, 64, 65]. Their common approach involves manually extracting urban entities and defining urban relations to construct an urban knowledge graph. For example, [6] construct a dedicated spatiotemporal knowledge graph regarding trajectory and timestamp as entities to improve trajectory prediction and [7] construct user check-in relations to help mobility prediction. Nevertheless, existing UrbanKGs heavily rely on manual design, leading to high labor costs.

## 7 Conclusion

In this work, we proposed UrbanKGent, the first automatic UrbanKG construction agent framework with LLMs. We first constructed a knowledgeable instruction set to adopt LLMs for different UrbanKGC tasks. Then, we proposed a tool-augmented iterative trajectory refinement module to facilitate the instruction tuning of various large language models. Extensive experimental results demonstrate the advancement of UrbanKGent in improving UrbanKGC tasks. The obtained UrbanKGent agent family, consisting of 7/8/13B version, with lower latency and cost compared with derving GPT-4 for UrbanKG construction. We hope the open-source UrbanKGent can foster future urban knowledge graph research and broader smart city applications.

## Acknowledgments and Disclosure of Funding

This work was supported by the National Natural Science Foundation of China (Grant No.62102110, No.92370204), National Key R&D Program of China (Grant No.2023YFF0725004), Guangzhou-HKUST(GZ) Joint Funding Program (Grant No.2023A03J0008), Education Bureau of Guangzhou Municipality.

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

Table 5: The detailed statistic of RTE datasets. We report the maximum length, minimum length, and average length of urban text in the RTE dataset.

| Dataset | Max Length | Min Length | Avg Length | # Records |
|---|---|---|---|---|
| NYC-Instruct | 1,747 | 68 | 437 | 232 |
| CHI-Instruct | 1,120 | 25 | 408 | 122 |
| NYC | 2,708 | 51 | 433 | 2,089 |
| CHI | 1,883 | 32 | 445 | 1,102 |
| NYC-Large | 4,598 | 20 | 1,179 | 40,480 |
| CHI-Large | 4,597 | 36 | 825 | 28,868 |

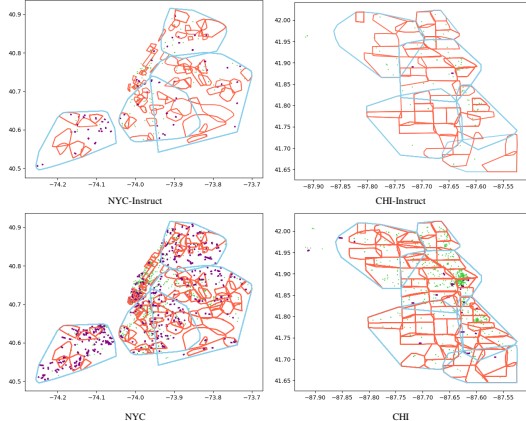

Figure 7: The geometry range visualization of the head entity and tail entity of four KGC datasets. The horizontal and vertical coordinates are longitude and latitude, respectively. The blue and red polygons stand for entities with the polygon geometry, the purple line string stands for the entities with linestring geometry and the green point is for the coordinate entities.

## A    UrbanKGC Construction

This section presents detailed UrbanKGC dataset statistic information for relational triplet extraction (RTE) and knowledge graph completion (KGC) tasks. Since two sequential tasks (i.e., RTE and KGC) of UrbanKGC are within an open-world setting (i.e., no predefined ontology) [40, 41]. Therefore, for the RTE task, every data record is an urban text without the triplet label. For the KGC task, every data record is a quadruple (i.e., head entity name, head entity geometry, tail entity name, tail entity geometry) without the geospatial relation label.

**RTE Dataset.** To facilitate the understanding of the constructed RTE dataset, we summarize the distribution of urban textual corpus in the six RTE datasets. As shown in Table 5, the entity

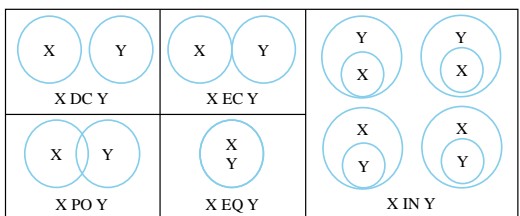

Figure 6: Given two geo-entities X and Y, the illustrative visualization of five types of RCC relationships. In this work, we consider entities with coordinate geometry to a circles with very small radii.

distribution and textual statistics of the instruct dataset and test data are similar. In addition, the record in dataset (i.e., NYC-Instruct and CHI-Instruct) used for instruction tuning is not overlapping with that in test dataset (i.e., NYC and CHI) and the real-world application dataset (i.e., NYC-Large and CHI-Large). This can avoid potential data leakage issues.

**KGC Dataset.** As for the KGC dataset, we provide illustrative visualization of the five RCC relationships [30] in Figure 6 for better understanding. Specifically, the disconnected (DC), externally connected (EC), partially overlapping (PO), equal (EQ), tangential and non-tangential proper parts

Table 6: The detailed geospatial tool name, description, interface input, and output. Each tool interface is implemented by python and it is self-programmed by GPT-4.

| Tool name | Tool description | Input | Output |
|---|---|---|---|
| Geohash | Geohash encoding | Geometry | Geohash code (8-bit) |
| Distance | Calculate the distance between two geo entities. | Geometry 1, Geometry 2 | Distance value (km) |
| Point2Polygon | Identify if a point belongs to a polygon | Point geometry, polygon geometry | True/False |
| Point4Linestring | Identify if a point intersects a linestring | Point geometry, linestring geometry | True/False |
| Linestring2Polygon | Identify if a linestring belongs to a polygon | Linestring geometry, polygon geometry | True/False |
| Linestring4Polygon | Identify if a linestring intersects a polygon | Linestring geometry, polygon geometry | True/False |
| Polygon2Polygon | Identify if a polygon belongs to a polygon | Polygon geometry, polygon geometry | True/False |
| Polygon4Polygon | Identify if a polygon intersects a polygon | Polygon geometry, polygon geometry | True/False |

(IN) together depict the basic geospatial relationship between urban entities. Moreover, to facilitate the understanding of the constructed KGC dataset, we visualize the geometry range of the head entity and the tail entity in four small KGC datasets. Due to the large amount of data and the display overlapping between entities and entities during visualization, we will not show the visualization results of NYC-Large and CHI-Large dataset, but the pattern is similar. As can be seen in Figure 7, the entity distribution in the instruct dataset (i.e., NYC-Instruct and CHI-Instruct) and test dataset (i.e., NYC and CHI) are similar. The pattern in NYC-Large and CHI-Large is similar. Due to the KGC task follows a zero-shot setting, we are unable to provide accurate distribution proportions of the 5 RCC relationships in the four datasets. However, it is intuitive can be seen that all four datasets contain the five RCC relationships shown in Figure 6, which provides a guarantee for the practical significance of KGC task in this work.

## B  Instruction Template

### B.1  Instruction Template in UrbanKGent

This section presents the detailed instruction template of the proposed UrbanKGent framework. Specifically, Figure 8(a-b) provides the detailed instruction template of knowledgeable instruction generation module and tool-based trajectory augmentation. The iterative trajectory self-refinement module is achieved by trajectory verifier and updater in Figure 8(c).

### B.2  Instruction Template in Baselines

We provide the detailed instruction template of all the baseline models in this work.

**LLMs-based Zero-shot Reasoning Methods.** We only provide task descriptions for the zero-shot reasoning method. The detailed instruction template is shown in Figure 9.

**LLMs-based In-context Learning Methods.** We first construct several few-shot demonstrations via the chain-of-thought prompting techniques [35], which is popular for automatic demonstration generation. The detailed instruction template is shown in Figure 9. Then, we add these demonstrations before the test question, and the detailed instruction template is shown in Figure 9.

**Vanilla Fine-tuning Methods.** We only provide task descriptions for vanilla fine-tuning methods without demonstrations. The detailed instruction template is shown in Figure 9.

**UrbanKGent Inference Methods.** These baseline models follow the same inference pipeline shown in Figure 8.

## C  Geospatial Toolkit

This work constructs a geospatial reasoning toolkit (eight interfaces in total in Table 6) by asking GPT-4 for self-programming. We obtained geospatial toolkit for point, linestring, and polygon geometry objects, supporting distance calculation and Geohash encoding for two geographical objects, and can also determine basic spatial relationships between two geometric objects, such as containment and intersection. For each interface, we provide its name and function description (e.g., *geohash encoding* in Table 6 and Figure 8b) into the instruction template.

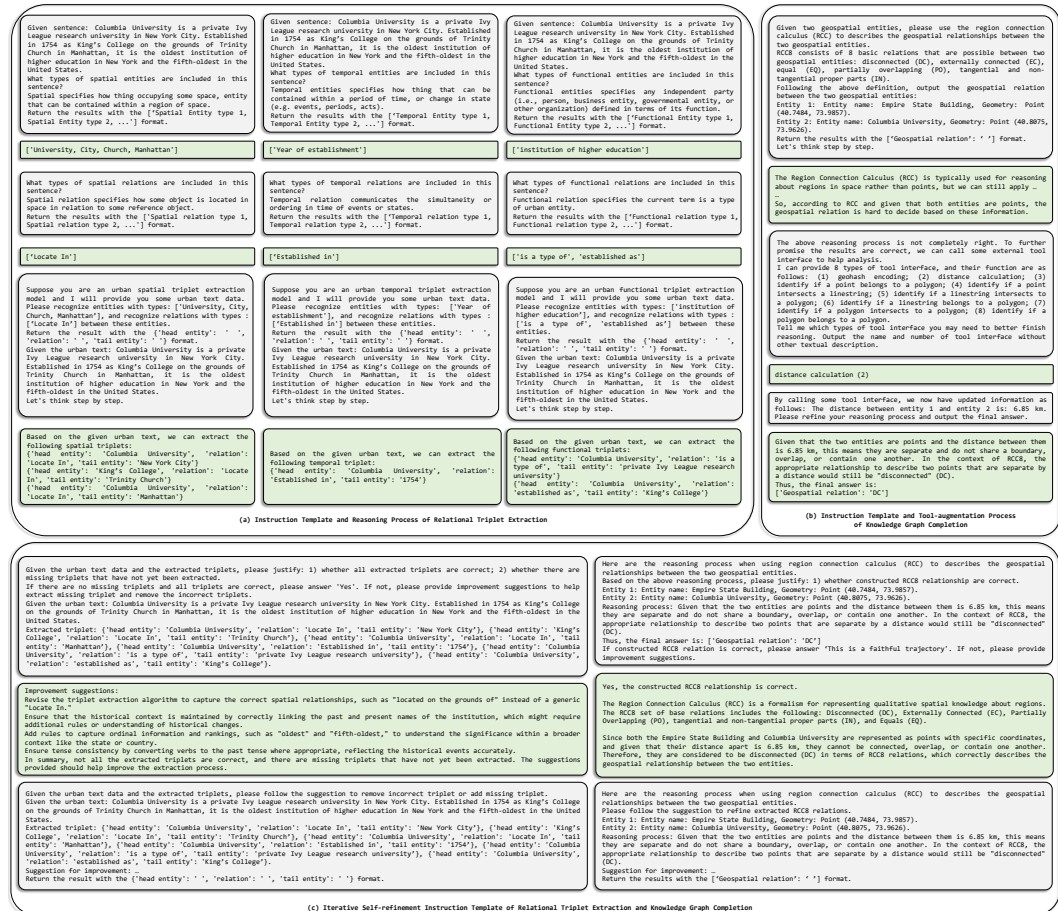

Figure 8: The reasoning process of UrbanKGent-13B and the detailed instruction template for UrbanKent inference pipeline. The content in the gray box is the instruction, and the content in the green box is the agent's response.

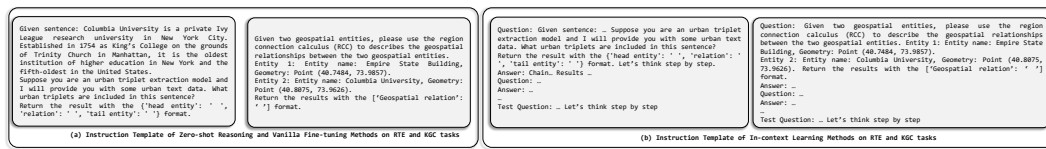

Figure 9: The instruction template of zero-shot reasoning, In-context learning and vanilla fine-tuning baselines.

Table 7: Illustrative RTE evaluation example when we utilize human evaluation and GPT evaluation method. We calculate the accuracy by counting the proportion of true triplets. The label for GPT evaluation method is invisible.

| Type | Urban text | Results | Label | Number of the true triplet | Number of the false triplet |
|------|-----------|---------|-------|---------------------------|-----------------------------|
| Human Evaluation | Columbia University is a private Ivy league research university in New York City. | \<Columbia University, locate-in, New York City \> | \<Columbia University, locate-in, New York City \> | 1 | 0 |
| GPT Evaluation | | | - | 1 | 0 |

# D Evaluation

This section presents the detailed evaluation process and examples to help the reader have a better understanding: **(1) Human Evaluation.** We employ human annotators to evaluate the results on 200 random samples. For the relational triplet extraction task, we first manually annotate the triplet label for each sample. Then, we manually evaluate the correctness of each triplet [40] based on annotation and calculate the accuracy value. For the knowledge graph completion task, we follow [11] to manually label the response as correct or wrong, and calculate the accuracy. **(2) GPT Evaluation.** Recently, many studies [52, 45] adopt LLM-based evaluation for open-domain tasks and empirically demonstrate that GPT-4's evaluation and human evaluation can be consistent [66]. In this work, we also use GPT-4 to evaluate the model performance on the full data to escape intensive labor. Specifically, given an UrbanKGC instruction and results, we prompt GPT-4 to return the confidence score and the justification (i.e., True/False), which will be further used to calculate the accuracy.

Moreover, we provide a comprehensive analysis to demonstrate why GPT and Human evaluations are highly aligned.

## D.1 Human Evaluation Process

For the relational triplet extraction (RTE) task, we provide an evaluation example in Table 7. Given the urban text sentence "Columbia University is a private Ivy league research university in New York City.", the human annotators are required to first label the triplet contained in this urban text, described as triplet *<Columbia University, locate-in, New York City>*. Based on the label, then, they are instructed to evaluate how many true triplets and false triplets in the results from the models. Finally, they will fill out the evaluation form (i.e., the number of the true triplets and the number of the false triplets). We will calculate the accuracy of results based on these annotated forms.

For the knowledge graph completion (KGC) task, given the head entity *<Columbia University, Point (40.8075, 73.9626)>* and the tail entity *<Empire State Building, Point (40.7484, 73.9857)>*, the human annotators are first required to complete their missing geospatial relationship from the five relation candidate (i.e., DC, EC, PO, EQ and IN). Specifically, the annotation could be achieved by manually visualizing the location of two entities given on the map and following the RCC relation rule in Figure 6 to determine their geographical relationships. Finally, they will fill out the evaluation results (i.e., True/False). We will calculate the accuracy based on these evaluation results.

## D.2 GPT Evaluation Process

For the relational triplet extraction (RTE) task, we provide an evaluation example in Table 7. Given the urban text sentence "Columbia University is a private Ivy league research university in New York City.", and the model extraction results (i.e., *<Columbia University, locate-in, New York City>*), we directly prompt GPT-4 to fill out the evaluation form (i.e., the number of the true triplets and the number of the false triplets). Then, the accuracy could be obtained based on these self-evaluated results.

For the knowledge graph completion (KGC) task, given the head entity *<Columbia University, Point (40.8075, 73.9626)>* and the tail entity *<Empire State Building, Point (40.7484, 73.9857)>* and completed geospatial relationship, we directly prompt GPT-4 the justify if the results are correct. Specifically, we will first explicitly call eight external geospatial tools in Table 6, and combine all these eight calculation results into the prompt. Then we feed these prompts into GPT-4 to help it

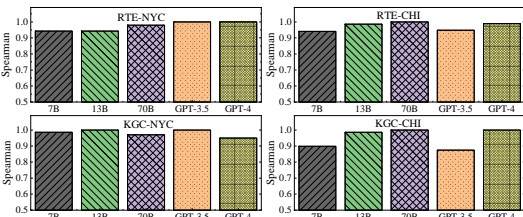

Figure 10: The Spearman correlation between the GPT evaluation and human's evaluation under five different LLM backbones (i.e., Llama-2-7B, Llama-2-13B, Llama-2-70B, GPT-3.5 and GPT-4).

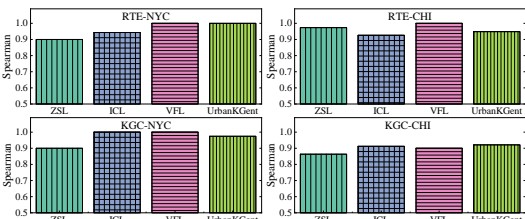

Figure 11: The Spearman correlation between the GPT evaluation and human's evaluation under four LLM paradigms (i.e., Zero-shot learning (ZSL), In-context learning (ICL), Vanilla fine-tuning (VFL) and UrbanKGent inference).

make a comparable accurate evaluation. Finally, we obtain the evaluation results (i.e., True/False) for every record. We will calculate the accuracy based on these evaluation results.

However, such an evaluation process carries risks as we cannot guarantee the infallibility of GPT-4. To ensure the validity of the evaluation, we have conducted extensive experiments to demonstrate that the GPT evaluation method aligns well with human evaluation.

### D.3 Evaluation Consistency

We conduct experiments to assess the consistency between GPT-4's evaluation results and human evaluation results. In this work, we follow GPTScore [67] to use Spearman coefficient [68] to investigate the correlation between GPT evaluation and human evaluation. Based on the evaluation results in Tabel 3, we can conduct correlation analysis separately by the types of model and paradigm.

**Comparison of LLM Backbones.** As shown in Figure 10, we observe that the Spearman correlation coefficients on all LLM backbones are greater than $0.8$, which means the human evaluation and GPT evaluation on these five LLM backbones is highly correlated.

**Comparison of LLM Paradigm.** Moreover, we also conduct correlation under different LLM paradigms for UrbanKGC tasks. As can be seen in Figure 11, all four types of paradigm have a higher Spearman value above $0.85$. Therefore, deriving the GPT evaluation method is also applicable to different LLM paradigms when finishing UrbanKGC tasks as its evaluation aligns well with that of the human being.

**Analysis of Evaluation Repeat.** Nevertheless, the above correlation analysis is based on results in Table 3, which are obtained after only single evaluation. To deeply understand the evaluation mechanism, We repeatedly instruct GPT-4 to generate 1, 3, 5, and 10 evaluation results to understand the variance of the evaluation method. To save the cost, we perform the above analysis using UrbanKGent-7B only on the NYC dataset. The results are shown in Table 8, although the coefficient value will decrease as the number of repeated experiments increases. However its value has always remained within a high correlation range (roughly $0.85$).

## E    In-Depth Analysis

In this section, we aim to perform more comprehensive analysis of our proposed UrbanKGent agent and hope to answer the following research questions. **(1) RQ1**: How does the constructed

Table 8: The average Spearman correlation value between human evaluations and GPT-4 evaluations. "Repeat X times" refers to instructing GPT-4 to generate judgments X times, and adopting the answer that appears most frequently (e.g., True/False for the KGC task and Number of the true triplet for the RTE task) as the final decision.

| Task | Repeat Once | Repeat 3 times | Repeat 5 times | Repeat 10 times |
|------|-------------|----------------|----------------|-----------------|
| RTE  | 98.56%      | 87.31%         | 86.42%         | 85.27%          |
| KGC  | 94.86%      | 91.23%         | 90.09%         | 91.24%          |

Table 9: The experimental results of relational triplet extraction (RTE) and knowledge graph completion (KGC) on the NYC-Large dataset and CHI-Large dataset. To save the cost, We choose the best approaches in zero-shot-learning (ZSL), In-context-learning (ICL), Vanilla Fine-tuning (VFT), and UrbanKGent Inference setting.

| Models | NYC-Large | | | | CHI-Large | | | |
|--------|-----------|--|--|--|-----------|--|--|--|
| | GPT(acc/confidence) | | Human(acc) | | GPT(acc/confidence) | | Human(acc) | |
| | RTE | KGC | RTE | KGC | RTE | KGC | RTE | KGC |
| ZSL | 0.36/4.01 | 0.45/3.95 | 0.42 | 0.31 | 0.38/4.27 | 0.33/3.67 | 0.44 | 0.36 |
| ICL | 0.39/4.36 | 0.48/4.15 | 0.48 | 0.39 | 0.38/4.22 | 0.35/3.53 | 0.40 | 0.35 |
| VFT | 0.37/4.05 | 0.46/3.98 | 0.45 | 0.35 | 0.33/4.25 | 0.28/3.67 | 0.38 | 0.34 |
| UrbanKGent Inference | 0.43/4.13 | 0.51/3.87 | 0.51 | 0.43 | 0.47/4.33 | 0.43/3.63 | 0.49 | 0.42 |
| UrbanKGent-7B | 0.44/4.27 | 0.50/4.07 | 0.53 | 0.44 | 0.48/4.25 | 0.42/3.85 | 0.55 | 0. 46 |
| UrbanKGent-8B | 0.43/4.07 | 0.51/4.16 | 0.52 | 0.44 | 0.49/4.21 | 0.43/3.77 | 0.53 | 0. 44 |
| UrbanKGent-13B | 0.46/4.56 | 0.52/3.67 | 0.55 | 0.46 | 0.55/4.29 | 0.49/3.24 | 0.58 | 0.49 |

UrbanKGent perform compared with existing paradigms on larger real-world dataset? **(2) RQ2**: How do different components (e.g., the knowledgeable instruction generation) affect the performance? **(3) RQ3**: How the complexity and efficiency of proposed UrbanKGent framework? **(4) RQ4**: How about the UrbanKGent inference trajectories when completing UrbanKGC tasks? **(5) RQ5**: How can the constructed UrbanKGent provide application service to real-world scenarios?

## E.1   RQ1: Evaluation on Larger Dataset

As shown in Table 9, we derive UrbanKGent-13B for urban knowledge graph construction using constructed large-scale dataset in NYC and CHI. Specifically, we directly use the text record in NYC-Large and CHI-Large for the relational triplet extraction task. Then, we randomly sample the head-tail entity pairs (both of head and tail entities contain geometry information) from these triplets for knowledge graph completion. Nevertheless, iterating all head-tail entity pairs is a time-consuming task, so we just construct a KGC dataset consistent with the sacle of the RTE dataset. By performing RTE and KGC tasks, we obtain two large-scale UrbanKGs shown in Table 4. Compared with existing dataset and benchmark UUKG [8], we can clearly observe that our agent can only use one-fifth data to construct the UrbanKG with the same scale entities and triplets, but extend the relationship types to a thousand times. It is worth noting that all construction process is completed by a LLM agent without any mannul effort. We think it is the core of this work.

Moreover, we also report the performance of the RTE and KGC in NYC-Large dataset. To save the cost of GPT self-evaluation, we only choose the best approaches in ZSL, ICL, VFT and UrbanKGent Inference. The experimental results are shown in Table 9. As can be seen, the fine-tuned UrbanKGC agent, whether 7B or 13B version, could achieve state-of-art performance on UrbanKGC tasks.

## E.2   RQ2: Ablation Studies

We conduct an in-depth analysis of the proposed instruction generation and tool-augmented iterative trajectory refinement module on the NYC dataset. Specifically, for the RTE and KGC task, we validate the effectiveness of each block by comparing the following variants: (1) UrbanKGent-7B♠ removes knowledgeable instruction template in RTE and KGC task; (2) UKGent* removes multi-view design in RTE task; (3) UrbanKGent-7B‡ removes external geospatial tool invocation block; (4)

Table 10: Effect of different blocks.

| Models | GPT (acc/confidence) | | Human (acc) | |
|---|---|---|---|---|
| | RTE | KGC | RTE | KGC |
| UrbanKGent-7B♠ | 0.38/4.17 | 0.42/3.98 | 0.37 | 0.34 |
| UrbanKGent-7B⋆ | 0.34/4.06 | 0.45/4.02 | 0.34 | 0.39 |
| UrbanKGent-7B‡ | 0.45/4.32 | 0.40/3.97 | 0.45 | 0.23 |
| UrbanKGent-7B† | 0.44/4.10 | 0.47/3.85 | 0.46 | 0.43 |

Table 11: Comparison among LLM-based UrbanKGC methods in four ways.

| Method | Extra Knowledge | Require Fine-tuning | Tool Invokation | Self Refinement |
|---|---|---|---|---|
| ZSL | × | × | × | × |
| ICL | √ | × | × | × |
| VFT | √ | √ | × | × |
| UrbanKGent Inference | √ | × | √ | √ |
| UrbanKGent | √ | √ | √ | √ |

UrbanKGent-7B† removes iterative trajectory self-refinement. We summarize the results in Table 10, and obtain the following observations.

First, knowledgeable instruction generation contributes to the overall performance of both RTE and KGC tasks. We observe a performance degradation by removing the knowledgeable instruction template. Second, the multi-view instruction design provides the most performance gain, which matches our intuition that the UrbanKG text contains heterogeneous relationships that can be effectively extracted by multi-view prompting design. Third, the tool invocation is very important for the KGC task, as we can observe significant performance degradation after removing the tool invocation. In addition, the iterative trajectory self-refinement brings consistent performance gain for both the RTE and KGC tasks.

### E.3 RQ3: Complexity and Effiency Analysis

We make a comparison with the four paradigms to demonstrate the advantages of the constructed agent, which is shown in Table 11. Compared with Zero-shot reasoning (ZSL), In-context Learning (ICL), Vanilla Fine-tuning (VFT), and UrbanKGent Inference, UrbanKGent can incorporate extra urban knowledge, invoke external tools and iteratively self-refine to help better complete UrbanKGC tasks.

Moreover, we also provide comprehensive efficiency analysis to show the latency and cost of different models when completing UrbanKGC tasks. Specifically, we report the total inference time and cost[9] of each method completing with 1,000 RTE and KGC tasks. For the cost of GPT-4 service, we first count the number of prompt token and completion token spent on 1,000 RTE and KGC tasks, and then calculate the cost based on billing standards. As for the cost of UrbanKGent-13B, we first count the GPU running time spent on 1,000 RTE and KGC tasks, and then calculate the cost based on the A800 charging standard.

In addition, as reported in table 12, we also provide detailed inference latency of UrbanKGent family when deriving them for constructing different-scale of UrbanKGs.

### E.4 RQ4: Case Study

As shown in Figure 8, we present the detailed reasoning process of constructed UrbanKGent-13B when finishing urban relational triplet extraction and knowledge graph completion task. Since the

---

[9]We subscribe to the NVIDIA A800 computing resources and GPT service from HKUST(GZ). Following the standard price instruction, we could calculate cost of GPT-based baselines or Llama-based baselines.

Table 12: The inference latency comparison of UrbanKGC using UrbanKGent family. We use two middle-size dataset (i.e., NYC and CHI) and two large-scale dataset (i.e., NYC-Large and CHI-Large) for UrbanKG construction.

| Dataset | Latency (minutes) | | | Data Volume |
| --- | --- | --- | --- | --- |
| | UrbanKGent-7B | UrbanKGent-8B | UrbanKGent-13B | |
| NYC | 1.19 | 1.58 | 3.19 | 2,089 |
| CHI | 0.62 | 1.13 | 1.68 | 1,102 |
| NYC-Large | 23.07 | 30.76 | 61.93 | 40,480 |
| CHI-Large | 16.55 | 21.93 | 44.47 | 28,868 |
| - | 0.57 | 0.76 | 1.53 | Per 1,000 records |

Table 13: The statistic of entity and relation ontology of constructed UrbanKGs on NYC-Large and CHI-Large datset.

| UrbanKG Dataset | # Coarse-grained Entity Ontology | # Fine-grained Entity Ontology | # Coarse-grained Relation Ontology | # Fine-grained Relation Ontology | # Entity | # Triplet |
| --- | --- | --- | --- | --- | --- | --- |
| NYC-Large | 4 | 6,281 | 4 | 2,138 | 228,928 | 905,442 |
| CHI-Large | 4 | 2,559 | 4 | 1,336 | 95,813 | 563,290 |

iterative self-refinement process contains excessive text, we display the reasoning process of only one iteration.

### E.5 RQ5: Agent Application

We have released the UrbanKGent family consisting of 7B, 8B and 13B version in the Huggingface. The opensourced UrbanKGent family offer urban knowledge graph construction service for the researcher in this field. We provide application example in New York and Chicago. Specifically, following the UrbanKGent Inference framework, we sequentially derive UrbanKGent-13B for relational triplet extraction and knowledge graph completion. The obtained initial UrbanKGs encodes diverse urban spatial, temporal and functional knowledge. Then, we propose to use a two-stage triplet filtering and relation merging operation to further improve the quality of constructed UrbanKGs.

In the first stage, low-frequency relations (occur 5 times or less) are merged into high-frequency relations based on relation similarity threshold. The remaining low-frequency triples, whose similarity with any high-frequency relation is below the threshold will be filtered out. In the stage two, we first perform relation clustering based on the embedding of relations. Then, Within each cluster, we prompt LLM to identify semantically similar relations that can be merged into a single relation category, resulting the final set of merged relations.

As shown in Table 13, both urban entity and relation can be pre-categorized into 4 coarse-grained ontologies: spatial, temporal, functional, and others. The entity percentage of spatial, temporal, functional and others in NYC-Large is (68.34%, 17.66%, 10.39%, 3.61%), and in CHI-Large is (63.29%, 16.68%, 12.09%, 7.94%). The relation percentage of spatial, temporal, functional and others in NYC-Large and CHI-Large are (57.36%, 16.38%, 20.89%, 5.37%) and (60.57%, 15.11%, 21.67% and 2.65%), respectively. After multi-view entity recognition and relation extraction (shown in Figure 4(a) in our paper), the fine-grained entity ontologies (6,281 and 2,559 entity types of NYC-Large and CHI-Large UrbanKGs, respectively) and fine-grained relation ontologies (2,138 and 1,366 relation types of NYC-Large and CHI-Large UrbanKGs, respectively) are obtained.

## F Limitation and Future Work

This work has limitation on the further application demonstration of construction UrbanKGs, although proposed UrbanKGent family could construct a UrbanKG with a hundreds relationship using only one-fifth of data. In addition, the evaluation method in this work is cost-intensive although GPT evaluation and Human evaluation has been experimentally demonstrated to be consistent. Despite the above limitations, we hope the opensource UrbanKGC agent can foster more extensive UrbanKG research and broad smart city application. In the future, we will derive extra image-modality data to further enrich UrbanKGC.

