# OpenReview forum: "UrbanKGent: A Unified Large Language Model Agent Framework for Urban Knowledge Graph Construction"
_NeurIPS.cc/2024/Conference — NeurIPS 2024 poster_

### Official Review · Reviewer_e3YZ · 2024-07-10

**Soundness:** 3
**Presentation:** 2
**Contribution:** 2
**Rating:** 6
**Confidence:** 4

**Summary:**

This paper presents a framework denoted UrbanKGent, for finetuning an LLM to assist the construction of knowledge graph construction (specifically triplet extraction relation prediction). The gist of the study is to construct an ad-hoc corpus for finetuning. In this process, a method of trajectory refinement is proposed to enhance both the effectiveness and explainability of the framework.

**Strengths:**

1) An interesting and important application domain of LLM
2) A generally well-developed workflow for tackling the target problem
3) The evaluation is more or less adequate for showing the superiority

**Weaknesses:**

1) It seems that the generation of the corpus, which is a key ingredient of the framework, lacks a systematic and scientific methodology for its generation.
2) The transferability is questionable – is this method useful in cities that are not finetuned?
3) The description of the roles of GPT4 and the smaller LLM that is finetuned is vague.

**Questions:**

See weaknesses.

**Limitations:**

See weaknesses.

---

> ### Author Rebuttal · Authors · 2024-08-07
>
> > [W1] It seems that the generation of the corpus, which is a key ingredient of the framework, lacks a systematic and scientific methodology for its generation.
>
> Sorry for the confusion. As mentioned by the reviewer, the corpus is very important for the UrbanKGent framework. Therefore, we construct the corpus by uniform sampling (Line 308-310) from multi-source urban data (i.e., AOI, road network, POI, review, and web page), which are widely used for urban knowledge distillation. By finetuning with the generated corpus, we demonstrate the performance of LLMs on UrbanKGC tasks could be largely improved. We will provide a more clear description of the corpus generation in our final paper version to avoid potential confusion.
>
> > [W2] The transferability is questionable – is this method useful in cities that are not finetuned?
>
> Thanks to the reviewer's question to help us clarify potential confusion. The proposed UrbanKGent framework could enhance UrbanKGC performance in both fine-tuning and non-fine-tuning scenarios. Specifically, as demonstrated in Table 3 of our paper, the performance of various LLM backbones consistently improves only using the UrbanKGent Inference pipeline without fine-tuning, compared with zero-shot reasoning and In-context-learning paradigms. Therefore, in cities where the LLM has not been fine-tuned, our method retains practical usage value.
>
> > [W3] The description of the roles of GPT4 and the smaller LLM that is finetuned is vague.
>
> Sorry for the confusion. As mentioned in Section 4.3 (Line 286-288) in our paper, we derive GPT-4 for trajectory generation consisting of two steps (i.e., the instruction generation and iterative trajectory refinement shown in Figure 4). The obtained trajectory will be further used to fine-tune the small LLMs. The fine-tuned smaller LLMs then will be used for UrbanKGC tasks with faster inference speed and lower cost.

---

> > ### Comment · Reviewer_e3YZ · 2024-08-11
> > **Thanks**
> >
> > I would like to thank the authors for the response. I would like to keep my rating as I find the argument about transferability is not that convincing.

---

> > > ### Author Response · Authors · 2024-08-11
> > > **Response to e3YZ**
> > >
> > > Dear Reviewer e3YZ:
> > >
> > > We are pleased that the previous rebuttal clarified most of the confusion. However, we apologize for the remaining confusion regarding "the transferability of the proposed framework in cities that are not fine-tuned." Please check the following response for a more detailed explanation.
> > >
> > > We agree with the reviewer that it is crucial to assess whether the proposed UrbanKGent framework can enhance UrbanKG construction in cities where the data was not used for fine-tuning. In fact, we have discussed this scenario in our main results. Specifically, we **directly apply the proposed UrbanKGent framework (i.e., UrbanKGent Inference in Table 3)** to different LLMs without fine-tuning, and observe the performance changes compared to prompting LLMs with zero-shot reasoning or in-context learning paradgim.
> > >
> > > As can be seen in Table 3, the proposed UrbanKGent framework was applied to the NYC and CHI datasets **without any fine-tuning of the LLMs for these specific cities.** Despite the absence of fine-tuning, the UrbanKGent framework could significantly improve the performance of various LLMs (including open-source models like Llama and Mistral, as well as API-based models like GPT-3.5 and GPT-4) on UrbanKGC tasks. For example, after applying the UrbanKGent framework, Llama-2-13B's RTE performance improved by 84.21%, rising from 0.19 to 0.35 under human evaluation, compared to the zero-shot reasoning paradigm. Similarly, Mistral-2-7B and GPT-3.5 showed improvements of 64.71% (0.17→0.28) and 38.71% (0.31→0.43), respectively. The improvements in KGC tasks are also substantial. These results demonstrate that our **proposed UrbanKGC framework can significantly enhance LLM performance on UrbanKGC tasks, even without city-specific data for fine-tuning**.
> > >
> > > We will include a more detailed description of the experimental settings and results analysis in the final version of our paper to avoid such confusion. We sincerely appreciate your insightful question and the opportunity to clarify this aspect of our work.
> > >
> > > Best,
> > >
> > > NeurIPS 2024 Conference Submission 16941 Authors

---

> > > > ### Comment · Reviewer_e3YZ · 2024-08-11
> > > > **Transferability**
> > > >
> > > > In this case the the transferability is indeed discussed. I have raised my rating.

---

### Official Review · Reviewer_rXu6 · 2024-07-11

**Soundness:** 3
**Presentation:** 4
**Contribution:** 3
**Rating:** 7
**Confidence:** 5

**Summary:**

The paper introduces UrbanKGent, a comprehensive framework utilizing a large language model (LLM) for constructing urban knowledge graphs (UrbanKGs). The key components of UrbanKGent involve creating an instruction set for tasks like relational triplet extraction and knowledge graph completion, which are tailored to urban data's heterogeneity and spatial characteristics. An iterative refinement module is employed to improve and adjust trajectories derived from the GPT-4 model, enhancing the quality of the knowledge graph. The framework is then fine-tuned using these enhanced trajectories on the Llama 2 and Llama 3 models, resulting in UrbanKGC agents, available in 7/8/13B versions.

**Strengths:**

1. Intensive efforts have been dedicated including crawling and preprocessing raw data, empirically validating the insufficiency of GPT-4, and instruction fine-tuning the UrbanKGC agent.
2. The paper focuses on real-world scenarios with domain knowledge infused in UrbanKGC agent construction and carries out comprehensive evaluations on real-world datasets using human and GPT-4 self-assessment.

**Weaknesses:**

1. The computational efficiency of the proposed algorithms and their scalability with the size of the dataset is not thoroughly discussed.

**Questions:**

1. As mentioned in Appendix Section D.1, human annotators are employed to fill out an evaluation form. Is it possible to provide more detailed information on human annotators, such as an exact number of human annotators?

**Limitations:**

As acknowledged in appendix section F, the paper has limitations on the further application demonstration of construction UrbanKGs and the evaluation method in this work is cost-intensive.

---

> ### Author Rebuttal · Authors · 2024-08-07
>
> > [W1] The computational efficiency of the proposed algorithms and their scalability with the size of the dataset is not thoroughly discussed.
>
> We sincerely appreciate the reviewer's valuable comment, which helps us improve the quality of our paper. As suggested, we provide detailed inference latency for the UrbanKGent family across different dataset scales in the following table. As shown in Table 1, by applying VLLM [1] techniques to accelerate our current framework, UrbanKGent-13B can automatically construct an UrbanKG with approximately one million triples (905,442 extracted from the NYC-Large dataset) in about 62 minutes. We also report the average inference time of the UrbanKGent family. For every 1,000 data records processed, the UrbanKGent takes 0.57 minutes on the 7B version, 0.76 minutes on the 8B version, and 1.53 minutes on the 13B version, respectively.
>
> Table 1: The inference latency comparison of UrbanKGC using the UrbanKGent family before and after using VLLM acceleration. We use two middle-size datasets (i.e., NYC and CHI) and two large-scale datasets (i.e., NYC-Large and CHI-Large) for UrbanKG construction. The accelerated inference latency is **bolded.**
>
> Dataset|Latency (minutes)|||Data Volume
> -|-|-|-|-
> || UrbanKGent-7B     | UrbanKGent-8B    |UrbanKGent-13B|
> NYC|6.83/**1.19**|7.62/**1.58**|16.48/**3.19**|2,089
> CHI|3.60/**0.62**|4.02/**1.13**|8.69/**1.68**|1,102
> NYC-Large|132.33/**23.07**|147.75/**30.76**|319.38/**61.93**|40,480
> CHI-Large|94.39/**16.55**|105.36/**21.93**|227.76/**44.47**|28,868
> -|3.27/**0.57**|3.65/**0.76**|7.89/**1.53**|Per 1,000 records
>
> Currently, the maximum dataset scale we used is about forty thousand. We are working on processing larger-scale data and analyzing potential scalability issues.
>
> **Reference**:
>
> [1] Kwon, Woosuk, et al. "Efficient memory management for large language model serving with pagedattention." *Proceedings of the 29th Symposium on Operating Systems Principles*. 2023.
>
> > [Q2] As mentioned in Appendix Section D.1, human annotators are employed to fill out an evaluation form. Is it possible to provide more detailed information on human annotators, such as an exact number of human annotators?
>
> Thanks to the reviewer's valuable suggestion. We invite 3 AI experts to fill out the evaluation form of the prediction results from various LLM variants. All of them possess work experience as algorithm engineers at Internet or AI companies. To avoid potential performance bias (as a priori, it is believed that larger-size LLMs are better), we do not reveal which results come from which LLMs. This rigorous process resulted in the annotation of 200 random samples. We will provide a more detailed description in Appendix Section D.1 in our final paper version. We sincerely appreciate the valuable suggestions provided by the reviewer.

---

> > ### Comment · Reviewer_rXu6 · 2024-08-14
> >
> > Thank you for addressing the concerns raised in the review. Your additional details regarding the computational efficiency and scalability of the UrbanKGent family are appreciated. The reported latencies and the application demonstrate a significant improvement in performance, particularly when dealing with larger datasets like NYC-Large and CHI-Large.
> >
> > Thank you again for your efforts to improve the paper. These additional clarifications will undoubtedly enhance the quality of the manuscript.

---

### Official Review · Reviewer_GYCX · 2024-07-12

**Soundness:** 2
**Presentation:** 3
**Contribution:** 2
**Rating:** 5
**Confidence:** 3

**Summary:**

The paper presents UrbanKGent, a unified large language model agent framework for urban knowledge graph construction. The framework consists of knowledgeable instruction generation, tool-augmented iterative trajectory refinement, and hybrid instruction fine-tuning. The authors evaluate the framework on two real-world datasets using both human and GPT-4 self-evaluation. The experimental results show that UrbanKGent outperforms 31 baselines in urban knowledge graph construction tasks and achieves state-of-the-art performance.

**Strengths:**

1. The paper proposes a unified framework for urban knowledge graph construction, addressing the challenges of heterogeneous relationship understanding and geospatial computing.

2. The knowledgeable instruction generation module and tool-augmented iterative trajectory refinement module are innovative and practical methodologies.

3. The experimental evaluation is comprehensive, including both human evaluation and GPT-4 self-evaluation.

4. The results show that UrbanKGent outperforms 31 baselines in urban knowledge graph construction tasks and achieves state-of-the-art performance.

**Weaknesses:**

1. This paper focuses on urban knowledge graph construction tasks. However, the details of the urban knowledge graph are not clearly defined. For example, the ontology of urban KGs, entities types, and relations types are not discussed. This limits the understanding of the proposed framework for readers who are not familiar with urban knowledge graphs.

2. The KGC tasks only focus on the geospatial relations. Is there any specific reason to select these relations? Can we predict other relations using the same framework?

3. The motivation of the KGC task is not clearly explained. If the geospatial information is given, can we directly use the tool to get their relations? The paper should provide more insights into the motivation behind KGC task and the necessity of using LLMs for it.

4. The ablation study is not clearly described. The paper should provide more details on the datasets, settings, and the performance of the final model used in Table 10. Otherwise, it is hard to evaluate the effectiveness of each module.

5. The iterative trajectory self-refinement module is not well explained. Based on what criteria is the trajectory refined? How many iterations are performed? How does the performance improvement as the number of iterations increases? Moreover, if LLM lacks the knowledge about urban KGs as the motivation of this paper, how can it be a good judger? Is there any analysis about the correctness of the comments and refined trajectory?

6. The cost for fine-tuning data construction is not discussed. How much data is required for fine-tuning the model? What is the cost of collecting and labeling the data for fine-tuning?

7. In experiments, the paper should compare UrbanKGent with existing UrbanKG construction methods rather than purely focusing on the LLMs which are not tailored for this task.

**Questions:**

1. What is the ontology of urban KGs?
2. What is the motivation for selecting geospatial relations for KGC tasks and using LLMs for this? Can we predict other relations using the same framework? Can we directly use the tool to get relations if geospatial information is given?
3. Please provide more analysis of the iterative trajectory self-refinement module.
4. Settings of the ablation studies.
5. Cost for fine-tuning data construction.
6. Can we compare UrbanKGent with existing UrbanKG construction methods?

---

> ### Author Rebuttal · Authors · 2024-08-07
>
> > W1&Q1
>
> Due to the tables related to weakness 1 and question 1 are included in the Supplementary PDF of the **Author Rebuttal**, we have moved the corresponding responses to the **Author Rebuttal** for better understanding. We apologize for any inconvenience this may cause.
>
> > W2,W3&Q2
>
> We choose geospatial relations for KGC tasks for two reasons. First, spatial relations are the majority in UrbanKG (about 60% in both datasets, as reported in Table 15 of PDF). Second, as validated by recent works [1-2], spatial relation understanding is one of the most challenging tasks for LLMs. Therefore, we use geospatial relations to demonstrate the effectiveness of the UrbanKGent in KGC. Other relations, including temporal and functional, can be predicted using this framework by extending corresponding instructions. For example, by injecting the time information (e.g., the date of building was built) of urban entities into the instruction, we can complete their missing temporal relation (e.g., built earlier than). The capability of UrbanKGent to identify other relation types is also echoed by its success in RTE tasks.
>
> Regarding external tool use, it is possible to invoke tools to calculate certain geospatial relations, but not all, with sufficient geospatial information. For example, given the Region Connection Calculus (RCC) of two entities, we can derive up to 5 defined geospatial relations using the GIS system. However, some spatial relations may not be able to be extracted using tools, especially when entity information is incomplete, or the corresponding relation type and tool are unknown. Overall, we acknowledge the merit of using external tools to derive urban relations. In fact, we have devised tool use as an indispensable block in the tool-augmented trajectory refinement block, and are working to incorporate more spatiotemporal tools to improve the effectiveness of our framework.
>
> Reference:
>
> [1] GeoLM. EMNLP. 2023. [2] Are Large Language Models Geospatially Knowledgeable? 2023.
>
> > W4&Q4
>
> We use UrbanKGent-7B as the final model, with the same setting in the overall experiment (Section 5.2) on the NYC dataset. For the ablation study (Line 720 - 724), we remove the knowledgeable instruction template ($UrbanKGent-7B^{\spadesuit }$), multi-view design ($UrbanKGent-7B^{\star }$), external geospatial tool invocation ($UrbanKGent-7B^{\ddagger}$), and iterative trajectory self-refinement ($UrbanKGent-7B^{\dagger}$) respectively from UrbanKGent-7B to validate their effectiveness. The results show that removing any of these modules will lead to performance degeneration, demonstrating their effectiveness. We will add more details in E.2 and provide more descriptions in the caption of Table 10.
>
> > W5&Q3
>
> We prompt the backbone LLM to automatically judge if a trajectory is faithful and provide refinement feedback for the unfaithful trajectory, more details are explained in **Lines 272-274**. Trajectory Updater will then follow the feedback to refine the current trajectory. For the maximum iterations, as explained in **Lines 280-282**, we set it to 3 to avoid excessive cost. To further understand the role of iteration numbers, we set maximum iterations from 0 to 10, and report the model's performance as well as the average stopping epochs in which the predefined stopping condition is satisfied (i.e., all trajectories are faithful).
>
> |The GPT-evaluation results on NYC dataset using UrbanKGent-7B||||
> -|-|-|-
> maximum Iteration|stopping epochs|RTE (acc)|KGC (acc)
> 0|-|0.44|0.47
> 3|2.46|0.46|0.49
> 5|2.84|0.47|0.48
> 10|3.25|0.48|0.49
>
> As can be seen, the model performance drops without iterations. However, larger than 3 iterations only lead to marginal improvements. This suggests that 3 iterations are cost-effective for the model to achieve the predefined stopping condition.
>
> Moreover, we agree with the reviewer that LLM cannot work as a good judger if it lacks urban knowledge. As shown in Table 10, the $UrbanKGent-7B^{\spadesuit }$, whose knowledgeable instruction template (Line 720-721) is removed, performs poorly, although it incorporates the iterative trajectory self-refinement method. Regarding the correctness of the refined trajectory, the performance improvement reported in the above table indicates that the refined trajectory is more accurate. The reported case study in Figure 8 in our paper also echos its effectiveness, e.g., the model can identify some missing triplets after self-refinement.
>
> > W6&Q5
>
> The constructed instruction dataset for fine-tuning can be found in **Supplementary Material**. As suggested, we provide statistics and the cost of prompting GPT-4 for instruction generation in the following Table:
>
> Task|Number of raw records|Number of instructions generated by UrbanKGent pipeline|Cost (dollar)
> -|-|-|-
> RTE|354|4,246|12.51
> KGC|354|2,011|18.07
> In total|708|6,257|30.58
>
> As can be seen, 6,257 instructions are used and the cost of calling GPT-4 is 30.58 dollars.
>
> > W7&Q6
>
> Existing UrbanKG construction methods rely heavily on manual extraction of urban entities and relations, as detailed in **Lines 394-400**. These manual methods, while effective, are not directly comparable to our automated approach without corresponding entities and relations annotated in the training data.
>
> However, we agree that a comparison can still provide valuable insights. To this end, we performed a quantitative comparison between previous manual methods and our proposed automatic paradigm using the latest UrbanKG benchmark [1]. As shown in **Table 4** in our paper, UrbanKGent can construct UrbanKGs with the same scale of triplets and entities in the benchmark by using only one-fifth of the data. Furthermore, our method expands the relationship types by a hundred times, demonstrating improvements in efficiency and comprehensiveness.
>
> This comparison underscores UrbanKGent's potential for automatic UrbanKG construction, making it a competitive alternative against manual methods.
>
> Reference:
>
> [1] UUKG. NeurIPS 2023.

---

> > ### Comment · Reviewer_GYCX · 2024-08-07
> >
> > Thanks for the responses, which partially addressed my concerns except for the necessity of using LLM for geospatial relation prediction. Thus, I raised my score to Borderline acceptance.

---

> > > ### Author Response · Authors · 2024-08-09
> > > **Response to Reviewer GYCX**
> > >
> > > Dear Reviewer GYCX:
> > >
> > > We are pleased that most of the confusion has been clarified by the previous Rebuttal. We appreciate your expertise and insights in helping us improve the quality of our paper. But we apologize for the difficulty the reviewer still experienced in understanding the necessity of using LLMs for geospatial relation prediction. Please check the following response for a more detailed explanation.
> > >
> > > We propose using LLMs for geospatial relation prediction for two primary reasons. First, existing GIS tools may struggle to extract certain spatial relations when geospatial information is incomplete. For instance, if the polygon data (i.e., the latitude and longitude boundaries) of two urban entities, such as Queens and Staten Island, is incomplete, traditional methods may fail to predict missing geospatial relations (e.g., whether they are disconnected). In contrast, LLMs can leverage both geospatial and semantic information to infer these relations. For example, an LLM might successfully infer that "Queens and Staten Island are geospatially disconnected" by directly using their semantics. Second, predicting new spatial relations often requires using multiple GIS tools or even developing new ones, which can be labor-intensive. LLMs, however, can efficiently manage this process by automatically routing tasks to existing tools or implicitly building a neural inference function for geospatial relation prediction. In our framework, we have designed the LLM to invoke various external tools to derive urban relations, and we are working to integrate more spatiotemporal tools to enhance the framework's effectiveness.
> > >
> > > Overall, the usage of LLMs for geospatial relation prediction has practical potential and has been explored in prior research. For example, recent studies [1-3] have quantitatively evaluated LLMs' ability to predict spatial relationships and perform certain spatial calculations. Furthermore, works like CityGPT [4], CityBench [5], and BB-GeoGPT [6] demonstrate the potential of LLMs in automating complex geospatial reasoning tasks, including geospatial relation prediction. We believe these efforts are crucial for the future deployment of LLM-based applications in urban and GIS contexts. In such a scenario, we present the first attempt to use LLMs for urban geospatial relation completion within the UrbanKG construction process. We believe our approach can serve as a valuable reference for researchers in this field.
> > >
> > > We will include the discussion on the necessity of using LLMs for geospatial relation completion in the final version of our paper. We sincerely appreciate your insightful question and the opportunity to clarify this aspect of our work.
> > >
> > > Best,
> > >
> > > NeurIPS 2024 Conference Submission 16941 Authors
> > >
> > >
> > > Reference:
> > >
> > > [1] Li, et al. “GeoLM: Empowering Language Models for Geospatially Grounded Language Understanding.” EMNLP. 2023.
> > >
> > > [2] Bhandari, et al. “Are Large Language Models Geospatially Knowledgeable?” ICAGIS. 2023.
> > >
> > > [3] Mooney, et al. “Towards Understanding the Geospatial Skills of ChatGPT: Taking a Geographic Information Systems (GIS) Exam.” SIGSPATIAL. 2023.
> > >
> > > [4] Feng, Jie, et al. "CityGPT: Empowering Urban Spatial Cognition of Large Language Models." *arXiv preprint arXiv:2406.13948* (2024).
> > >
> > > [5] Feng, Jie, et al. "CityBench: Evaluating the Capabilities of Large Language Model as World Model." *arXiv preprint arXiv:2406.13945* (2024).
> > >
> > > [6] Zhang, Yifan, et al. "BB-GeoGPT: A framework for learning a large language model for geographic information science." *Information Processing & Management* 61.5 (2024): 103808.

---

> > > > ### Author Response · Authors · 2024-08-12
> > > > **Response to Reviewer GYCX**
> > > >
> > > > Dear reviewer GYCX:
> > > >
> > > > We sincerely appreciate precious review time and valuable comments. We have carefully considered your comments and have provided corresponding responses, which we believe have covered your concerns. We hope to further discuss with you whether or not your concerns have been addressed. Please let us know if you still have any unclear parts of our work.
> > > >
> > > > Best,
> > > >
> > > > NeurIPS 2024 Conference Submission 16941 Authors

---

### Official Review · Reviewer_72X8 · 2024-07-12

**Soundness:** 3
**Presentation:** 2
**Contribution:** 3
**Rating:** 7
**Confidence:** 4

**Summary:**

This paper proposes a unified large-scale language model agent framework called UrbanKGent, specifically for the Construction of Urban Knowledge Graph Construction (UrbanKGC). UrbanKGent utilizes instruction generation of heterogeneous and geospatial information, as well as an iterative trajectory optimization module based on GPT-4, to further enhance the ability to extract critical knowledge from multi-source city data and effectively reduce the significant manual labor of traditional methods. By fine-tuning the hybrid instructions on the Llama 2 and Llama 3 series models, the researchers developed the UrbanKGC agent, including the UrbanKGent-7/8/13B version. Experiments on two real-world datasets show that the UrbanKGent family not only significantly outperforms 31 benchmark models on the UrbanKGC task, but is also more than 20 times more cost-effective than GPT-4, while being able to build a knowledge map of cities with hundreds of times richer relationships with less data.

**Strengths:**

1. UrbanKGent proposed in this paper provides a unified solution for building urban knowledge graphs, which automates the process of extracting key information from multi-source urban data and reduces the need for human intervention.

2. The authors propose instruction generation methods of "heterogeneity awareness" and "geospatial information fusion", which can better capture the characteristics of urban knowledge graph construction tasks and make up for the shortcomings of ordinary LLM in understanding complex heterogeneous relationships and geospatial computing capabilities.

3. Experimental results on two real-world data sets show that UrbanKGent not only significantly outperformed 31 benchmarks on the UrbanKGC task but also outperformed the state-of-the-art LLM GPT-4 by more than 10% at about 20 times lower cost, which has great potential for efficiency and economy in practical applications.

**Weaknesses:**

1. As the size of the dataset grows, the algorithmic efficiency and scalability of UrbanKGent may become an issue. Although not mentioned in the paper, in practical applications, the processing of large-scale data sets may require more computational resources and time. Although the paper notes limitations, the specifics may require more elaboration. For example, under what conditions may the model perform poorly and how these limitations affect the quality and reliability of the final knowledge graph?

2. UrbanKGent may face privacy and fairness challenges, especially when handling personal or sensitive information. I wonder what the author thinks about privacy and fairness. Although these issues are not discussed in detail in the paper, in actual deployment, if not properly addressed, they can lead to violations of personal privacy or unfairly affect specific groups.

3. The UrbanKGent framework is mainly aimed at urban knowledge graph construction tasks. I wonder if the author has considered the applicability of other types of knowledge graph construction tasks. Can the UrbanKGent framework proposed in this paper be further extended to the knowledge graph construction in other fields?

**Questions:**

NA

---

> ### Author Rebuttal · Authors · 2024-08-07
>
> > [W1] As the size of the dataset grows, the algorithmic efficiency and scalability of UrbanKGent may become an issue. Although not mentioned in the paper, in practical applications, the processing of large-scale data sets may require more computational resources and time. Although the paper notes limitations, the specifics may require more elaboration. For example, under what conditions may the model perform poorly and how these limitations affect the quality and reliability of the final knowledge graph?
>
> Thanks for the reviewer's insightful question, which helps us improve the quality of our paper. We agree with the reviewer that efficiency is important in practical agent deployment. As suggested, we provide a more detailed efficiency analysis of UrbanKGent in the following table, and we also use LLM acceleration techniques VLLM [1] to improve the scalability of UrbanKGent. As shown in Table 1, by applying VLLM techniques to accelerate our current framework, UrbanKGent-13B can automatically construct an UrbanKG with approximately one million triples (905,442 extracted from the NYC-Large dataset) in about 62 minutes. We also report the average inference time of the UrbanKGent family. As can be seen in Table 1, For every 1,000 data records processed, UrbanKGent takes 0.57, 0.76, and 1.53 minutes on 7B, 8B, and 13B versions, respectively.
>
> Table 1: Comparison of UrbanKGC inference latency (minutes) using the UrbanKGent family before and after using VLLM acceleration. We use two middle-size datasets (i.e., NYC and CHI) and two large-scale datasets (i.e., NYC-Large and CHI-Large) for UrbanKG construction. The accelerated inference latency is **bolded.**
>
> Dataset|Latency (minutes)|||Data Volume
> -|-|-|-|-
> || UrbanKGent-7B     | UrbanKGent-8B    |UrbanKGent-13B|
> NYC|6.83/**1.19**|7.62/**1.58**|16.48/**3.19**|2,089
> CHI|3.60/**0.62**|4.02/**1.13**|8.69/**1.68**|1,102
> NYC-Large|132.33/**23.07**|147.75/**30.76**|319.38/**61.93**|40,480
> CHI-Large|94.39/**16.55**|105.36/**21.93**|227.76/**44.47**|28,868
> -|3.27/**0.57**|3.65/**0.76**|7.89/**1.53**|Per 1,000 records
>
> Currently, the maximum dataset scale we used is about forty thousand. We are working on processing larger-scale data and analyzing the potential performance degradation and KG quality issues. Thanks to the reviewer for introducing these interesting research questions, and we will discuss them in the Section "Limitation and Future Work" in our paper.
>
> > [W2] UrbanKGent may face privacy and fairness challenges, especially when handling personal or sensitive information. I wonder what the author thinks about privacy and fairness. Although these issues are not discussed in detail in the paper, in actual deployment, if not properly addressed, they can lead to violations of personal privacy or unfairly affect specific groups.
>
> Thanks for the reviewer's insightful question. Currently, we collect raw urban data from various public data providers (e.g., OSM, Wikipedia) to construct UrbanKGent. Although sensitive information is not a critical issue in these open-source data, we agree with the reviewer's concern regarding potential data privacy and fairness issues once UrbanKGent is deployed. To address this, we can further perform safety alignment [1] in UrbanKGent to control its outputs. In addition, we can subscribe to external LLM services to filter the malicious information of UrbanKGent's output. The above two strategies have been widely used in many online LLM reasoning services (e.g., ChatGPT, Ernie Bot, and Tongyi Qianwen), and we believe they would be the practical solution. We will further discuss this in the "Limitations and Future Work" section of our paper to provide more insights.
>
> **Reference**:
>
> [1] Ji, Jiaming, et al. "Beavertails: Towards improved safety alignment of llm via a human-preference dataset." *Advances in Neural Information Processing Systems* 36 (2024).
>
> > [W3] The UrbanKGent framework is mainly aimed at urban knowledge graph construction tasks. I wonder if the author has considered the applicability of other types of knowledge graph construction tasks. Can the UrbanKGent framework proposed in this paper be further extended to the knowledge graph construction in other fields?
>
> Thanks for your insightful comment and question! Yes, although the proposed framework is designated to the urban domain, the construction pipeline can be easily extended to the construction of knowledge graphs for other domains. One of the most straightforward ways is to modify the instruction template to adapt this framework to other domains. For example, we can replace the urban knowledge currently encoded in the RTE instruction template with other domain knowledge, to adapt our framework for the knowledge graph construction in other fields.

---

> > ### Comment · Reviewer_72X8 · 2024-08-14
> > **Response**
> >
> > Thanks for your responses. MY concerns are addressed. I will raise my score to accept.

---

### Author Rebuttal · Authors · 2024-08-07

**Dear Reviewer GYCX**:

We thank you for the precious review time and valuable comments. To ease understanding, we have moved weakness 1 and question 1 here, which are related to UrbanKG entity and relation ontology statistics. Please refer to Table 13, Table 14, and Table 15 in the **PDF** for detailed information on UrbanKG entity and relation ontology statistics.

> [W1] This paper focuses on urban knowledge graph construction tasks. However, the details of the urban knowledge graph are not clearly defined. For example, the ontology of urban KGs, entity types, and relation types are not discussed. This limits the understanding of the proposed framework for readers who are not familiar with urban knowledge graphs.

> [ Q1] What is the ontology of urban KGs?

We apologize for the confusion. The constructed UrbanKGs can be found in **Supplementary Material**. As the reviewer suggested, we provide entity and relation ontology statistics in the PDF. As shown in Table 13, both urban entity and relation can be pre-categorized into 4 coarse-grained ontologies: spatial, temporal, functional, and others. After multi-view entity recognition and relation extraction (shown in Figure-4(a) in our paper), the fine-grained entity ontologies (1,028 and 755 entity types of NYC-Large and CHI-Large UrbanKGs, respectively) and fine-grained relation ontologies (2,138 and 1,366 relation types of NYC-Large and CHI-Large UrbanKGs, respectively) are obtained. To ease understanding, we also provide illustrative entity ontology (e.g., University) and relation ontology (e.g., Locate-in) examples in Table 14 and Table 15, respectively. In addition, we also display the entity and relation ontology distribution of constructed UrbanKGs on the NYC-Large and CHI-Large datasets. More detailed ontology information can be found in the Supplementary Material.

We will include detailed statistics about the UrbanKG ontology in the final version of our paper. We sincerely appreciate the reviewer's valuable suggestions.

Best,

NeurIPS 2024 Conference Submission 16941 Authors

---

### Decision · Program_Chairs · 2024-09-25

**Decision:**

Accept (poster)

**Comment:**

Existing Urban Knowledge Graph Construction (UrbanKGC, which encompass relation extraction and KG completion tasks) approaches either rely on manually designed patterns/rules or annotated corpus to train LLMs. This work proposes  an agent framework UrbanKGent to solve UrbanKGC with minimal human labor and has the following components:
1) prescripted LLM process prompts:
a) triple extraction prompts based on a two-turn dialog with LLM
b) KG completion prompts infused with geospatial information
c) toolkit description/invocation prompts
d) reasoning refinement prompts
2) iterative trajectory refinement based on a verifier/critique prompt and an updater prompt
3) to achieve efficient inference smaller LLMs are fine-tuning on the larger LLMs trajectories.

Evaluation on two real-world datasets using both human and GPT-4 self-evaluation show that UrbanKGent outperforms various LLM baselines and end-to-end models.

Reviewers recognize the importance of the application and the motivation behind the system design. This is solid work applying LLM to information extraction in the UrbanKG domain, and may set the stage for more advanced learning approaches in the future.